# Bioactive Terpenes from Marine Sponges and Their Associated Organisms

**DOI:** 10.3390/md23030096

**Published:** 2025-02-21

**Authors:** Yuan Yuan, Yu Lei, Muwu Xu, Bingxin Zhao, Shihai Xu

**Affiliations:** 1Department of Chemistry, College of Chemistry and Materials Science, Jinan University, Guangzhou 510632, China; yuany@stu2024.jnu.edu.cn (Y.Y.); 31000652@scau.edu.cn (Y.L.); 2Department of Epidemiology and Environmental Health, School of Public Health and Health Professions, University at Buffalo, Buffalo, NY 14260, USA; muwuxu@buffalo.edu

**Keywords:** marine sponges, sponge-derived microbe, metabolites, terpenes, biological activities

## Abstract

In recent years, marine natural products have continued to serve as a pivotal resource for novel drug discovery. Globally, the number of studies focusing on Porifera has been on the rise, underscoring their considerable importance and research value. Marine sponges are prolific producers of a vast array of bioactive compounds, including terpenes, alkaloids, peptides, and numerous secondary metabolites. Over the past fifteen years, a substantial number of sponge-derived terpenes have been identified, exhibiting extensive structural diversity and notable biological activities. These terpenes have been isolated from marine sponges or their associated symbiotic microorganisms, with several demonstrating multifaceted biological activities, such as anti-inflammatory, antibacterial, cytotoxic, anticancer, and antioxidant properties. In this review, we summarize 997 novel terpene metabolites, detailing their structures, sources, and activities, from January 2009 to December 2024. The structural features and structure-activity relationship (SAR) of different types of terpenes are broadly analyzed and summarized. This systematic and comprehensive review will contribute to the summary of and speculation on the taxonomy, activity profiles, and SAR of terpenes and the development of sponge-derived terpenes as potential lead drugs.

## 1. Introduction

Marine sponges represent the most primitive form of multicellular animals, characterized by a specialized luminal tubular structure. After more than 500 million years of evolution, marine sponges have developed a unique chemical and physical defense mechanism to adapt to the high-pressure, high-salted, anoxic, and lucifugal environment [1]. Therefore, they produce a wide range of valuable secondary metabolites with multiple biological activities. Growing evidence indicates that sponges harbor a wealth of symbiotic or epiphytic microorganisms in their bodies, which are likely to be the true producers of bioactive secondary metabolites [2]. In some cases, these microbial associates comprise as much as 40% of the sponge volume and can contribute significantly to host metabolism [3]. These microorganisms not only serve as food for sponges but also play a crucial role in forming the defensive mechanisms of sponges [4]. The mutually beneficial symbiotic relationship between sponges and microorganisms exhibits rich chemical diversity, which is important for obtaining symbiotic natural products of significant biotechnological value [5].

Indeed, sponges have great potential for discovering new drug candidates that can aid in the prevention and treatment of disease [6]. For example, Cytosar-U^®^ (Ara-C), a secondary metabolite derived from the Caribbean sponge *Cryptotheca crypta*, was approved by the United States Food and Drug Administration (FDA) for the treatment of leukemia in 1969, and it is still in use today for the treatment of acute lymphocytic leukemia (ALL) [7,8]. A large number of natural compounds are under preclinical investigation, and others are already on the market as drugs. Among the seven marine-derived drugs currently approved by the FDA, three are derived from sponges. In addition to Ara-C, which was mentioned earlier, there are vidarabine (Vira-A, Ara-A) and eribulin mesylate (Halaven^®^) [9]. However, none of them are terpenes. Nevertheless, some terpenoids derived from sponges have shown great potential for drug development in preclinical studies. Stellettin B is a marine triterpene isolated from the South China Sea sponge *Jaspis stellifera*. The research showed that stellettin B reduced the migration and invasion of hepatocellular carcinoma cells through reducing the activation of the MAPKs and FAK/PI3K/AKT/mTOR signaling pathways [10]. It also ameliorated collagen-induced arthritis in mice and had anti-glioblastoma cancer and anti-non-small cell lung cancer activities [11,12,13]. The chemical and biological investigation of marine sponges has consistently been one of the most active subfields within natural pharmaceutical discovery.

During the last fifteen years, it has been seen that sponge-associated terpenes have diverse chemical structures and a wide variety of biological activities (antimicrobial, antitumor, antioxidant, anti-inflammatory, antiviral, antimalarial, antitubercular, anti-aging, antifouling, and antiprotozoal) [14,15,16]. Previous reviews have only generalized to a particular class of sponges, or a particular class of terpenes, such as nitrogen-containing or triterpenes, and have covered a short period of time [17,18,19]; hence, the need for a comprehensive review that encompasses a substantial amount of data. This review collects all the related literature from January 2009 to December 2024 and completely describes the new sponge-associated terpenes from 249 articles. Figure 1 illustrates the percentage of different activities of new compounds discovered from January 2009 to December 2024. It is evident that cytotoxicity represents the biological activity with the most substantial proportion. Among the four primary terpenes contributing to this activity, sesterterpenes hold the largest share, followed by sesquiterpenes and diterpenes.

A similar trend is observed for the total number of active compounds in each terpene category during 2009–2024 (Figure 2). However, there is no doubt that one compound can only be classified as active or inactive. While they are classified into specific type of activity, they may be reclassified as having multiple activities.

Notably, with the increasing emphasis on, as well as the depth of research in, the field of marine sponges, the number of new sponge-derived compounds is on an overall downward trend. Similarly, the share of active compounds is also decreasing, especially reaching a low point in the years 2018–2020. However, in recent years, the number of new compounds has been on the rise with the advances in collection equipment and innovations in isolation methods, indicating that there is still a wealth of active compounds waiting to be discovered (Figure 3).

## 2. Methodology

The data collection for this research was conducted using the major online databases Web of Science and PubMed, employing the following descriptors: marine sponge, new terpene and terpene, and new terpenoid and terpenoid. The inclusion criteria were as follows: (1) a publication date from 1 January 2009 to 31 December 2024; (2) written in English; (3) terpenes were first found in the article. Initially, 496 results were identified (325 from PubMed and 171 from Web of Science). The exclusion criteria included the following: (1) book chapters and patents; (2) duplicate studies; (3) studies that were not within the scope; (4) terpenes that were not new. As a result of this survey, the articles included in this systematic review comprise 249 publications and 997 compounds (Figure 4).

## 3. Different Class of Terpenoids

### 3.1. Sesquiterpenes

The drimane-type sesquiterpene is a saturated naphthalene alkane with 15 carbons. Compounds **1**–**7** were isolated from the marine fungus *Aspergillus ustus* derived from the marine sponge *Suberites domuncula* [20]. Compounds **4** and **5** showed cytotoxic activities against the L5178Y cell line with EC_50_ values of 5.3 and 0.6 μg/mL, respectively. Compound **8** was obtained from an EtOAc extract of the marine-derived fungus *Aspergillus insuetus*, which was isolated from the Mediterranean sponge *Psammocinia* sp. [21]. Compound **8** showed mild cytotoxicity towards MOLT-4 human leukemia cells. Two new drimane-type terpenes (**9**–**10**) were isolated from the marine sponge *Hyrtios* sp. collected from Papua New Guinea [22]. In 2015, eight 4,9-friedodrimane-type sesquiterpenoids (**11**–**18**) were acquired from the marine sponges *Smenospongia aurea*, *Smenospongia cerebriformis*, and *Verongula rigida* collected from Key Largo, Florida [23]. Compounds **12** and the mixture of **13** and **14** started to exert antiproliferative activities at a concentration of more than 20 μM. Compound **18** inhibited tumor growth against the SW480 and HCT116 cell lines with IC_50_ values of 3.24 and 2.95 μM, respectively. Further investigation revealed that **18** displayed more potent cytotoxic activity on a suppressed *β*-catenin response in the transcription of (CRP)-positive cancer cell. Compounds **19**–**24**, which consisted of quinone (hydroquinone) and sesquiterpene moieties, were isolated from the marine sponge *Dysidea* sp. collected from the Federated States of Micronesia [24]. Notably, these compounds exhibited moderate to weak inhibition against Na^+^/K^+^-ATPase and cytotoxic activities towards the K562 and A549 cell lines. Three phenylspirodrimane-based meroterpenoids (**25**–**27**) with novel scaffolds were isolated from the sponge *Niphates recondite*-associated fungus *Stachybotrys chartarum* WGC-25C-6 [25]. Chartarolides A–C (**25**–**27**) exhibited remarkable cytotoxic activities against the HCT-116, HepG2, BGC-823, NCI-H1650, A2780, and MCF7 human tumor cell lines with IC_50_ values around 1.3 to 12.5 μM and showed strong inhibitory activities against the human tumor-related protein kinases of FGFR3, IGF1R, PDGFRb, and TrKB with IC_50_ values ranging from 2.6 to 20.3 μM.

Ebada and coworkers investigated a MeOH extract of the Indonesian marine sponge *Dactylospongia elegans*, afforded two undescribed drimane meroterpenoidal metabolites named dactylospongenones G and H (**28**–**29**) [26]. Yu and colleagues isolated ten sesquiterpenes (**30**–**39**) of different types from the sponge *Pseudoceratina purpurea* [27]. Pseudoceranoid D (**33**) showed weak cytotoxicity against the K562, H69AR, and MDAMB-231 cell lines with IC_50_ values of 3.01, 7.74, and 9.82 μM, respectively. Pseudoceranoid E (**34**) exhibited cytotoxicity against the H69AR cell line with an IC_50_ value of 2.85 μM. Daletos et al. had reported five new sesquiterpene aminoquinones (**40**–**44**) and two new sesquiterpene benzoxazoles (**45**–**46**) from the sponge *Dactylospongia metachromia* collected from Ambon, Indonesia [28]. Moreover, compounds **40**–**44** showed potent cytotoxicity against the mouse lymphoma cell line L5178Y with IC_50_ values ranging from 1.1 to 3.7 μM, and compounds **46** and **46** exhibited the strongest inhibitory activities against the ALK, FAK, IGF1-R, SRC, VEGF-R2, Aurora-B, and MET wt cell lines and NEK6 kinases (IC_50_ = 0.97–8.62 μM). Compounds **47**–**51** were isolated from the sponge *Dactylospongia elegans* collected in Palau and Malaysia [29]. The compounds **47** and **48** were found to activate HIF-1 and stimulate the expression of the HIF-1 target gene vascular endothelial growth factor (VEGF) in both T47D and PC-3 cells. Compounds **52**–**54** were isolated and characterized from the *Hyrtios* sp. collected from the South China Sea [30]. Hyrtiolacton A (**52**) is the first example of a pyrone-containing 4,9-friedodrimane-type sesquiterpene. Nakijinol G (**54**) showed PTP1B inhibitory activity with IC_50_ values of 4.8 μM. Eleven new nitrogenous meroterpenoids (**55**–**65**) were isolated from the marine sponge *Dysidea cinerea* collected in the South China Sea [31]. Compounds **55**–**57** and **60** exhibited moderate inhibitory activities against PTP1B, ATP-citrate lyase, and SH2 domain-containing phosphatase-1 with IC_50_ values of 2.8–27 μM.

The chemical structures of compounds **1**–**65** are depicted in Figure 5, while the remaining information, including names and marine sources, is presented in Appendix A.

Bisabolane-type sesquiterpenoids, a class of monocyclic sesquiterpenoids, are widely distributed in nature and have a variety of biological activities. Compounds **66**–**68** were isolated from the marine sponge *Cacospongia mycofijiensis* collected from Papua New Guinea [32]. Two of them (**66** and **67**) showed moderate inhibition against the parasite *Trypanosoma brucei*, which is responsible for sleeping sickness, with IC_50_ values of 6 and 16 μg/mL, respectively. Compound **69** had been isolated from the sponge *Phycopsis* sp. collected from the Mandapam coast in India [33]. Two new uncommon bisabolene-type sesquiterpenes (**70**–**71**) were isolated from the sponge *Axinyssa* sp. off the Lingshui Bay, Hainan Province, China [34]. Compounds **72**–**75** were isolated from the fungus *Aspergillus* sp., which was obtained from the sponge *Xestospongia testudinaria* collected from the South China Sea [35]. The antibacterial activity and in vitro cytotoxic activity tests revealed that **72**–**75** showed selective inhibitory activities on six pathogenic bacteria—*S. albus*, *B. subtilis*, *B. cereus*, *S. lutea*, *E. coli*, and *M. tetragenus*—and two marine bacterial strains, *V. Parahaemolyticus* and *V. anguillarum*. Moreover, **75** strongly inhibited larval settlement at a concentration of 25 μg/mL. Compounds **72**–**75** were also found to have weak-to-no inhibitory activity against the HL-60 and A-549 cell lines and acetylcholinesterase. The other three compounds, **76**–**78**, were isolated from the same fungus and tested for the same activity [36]. The tests revealed that **76**–**78** showed selective inhibitory activities on all strains. In addition, **76** and **78** exhibited moderate cytotoxicity towards the HepG-2 human hepatoma cell line and Caski human cervical cell line.

An investigation of a Formosan sponge *Axinyssa* sp. collected from coral reefs off the coast of Pingtung in Taiwan, China, has led to the isolation of two nitrogenous bisabolene-type sesquiterpenes (**79**–**80**) [37]. Compound **79** exhibited moderate-to-weak cytotoxicity against the Molt4 and K562 cancer cell lines with IC_50_ values of 14.3 and 4.7 μg/mL, respectively. In 2016, a chemical investigation of the marine sponge *Axinyssa variabilis* yielded two new highly oxidized formamide bisabolene sesquiterpenes (**81**–**82**) [38]. A new rearranged nitrogenous bisabolene-type sesquiterpene, halichonic acid B (**83**), was obtained from the marine sponge *Axinyssa* sp. collected from the same place [39]. Wang and colleagues isolated six new bisabolane-type phenolic sesquiterpenoids, including plakordiols A-D (**84**–**87**), (7*R*, 10*R*)-hydroxycurcudiol (**88**), and (7*R*, 10*S*)-hydroxycurcudiol (**89**) from the marine sponge *Plakortis simplex* collected from the South China Sea [40]. Compound **88** inhibited bone morphogenic protein (BMP)-induced alkaline phosphatase activity in mutant BMP receptor-carrying C2C12 cells with IC_50_ values of 51 μM. The chemical structures of compounds **66**–**89** are depicted in Figure 6.

Sesquiterpene quinone/quinols are characterized by a C_15_-sesquiterpenoid unit incorporating a C6-benzoquinone/quinol moiety [41]. In 2009, a new sesquiterpene aminoquinone named dysideamine (**90**) was obtained from the Indonesian marine sponge *Dysidea* sp. [42]. Compound **90** showed a neuroprotective effect against iodoacetic acid (IAA)-induced HT22 cell death at a concentration of 10 μM. Researchers from the Shanghai Institute of Materia Medica reported a new sesquiterpene quinone (**91**), which was isolated from the sponge *Dysidea villos* [43]. Compound **91** showed moderate PTP1B inhibitory activity and cytotoxicity with IC_50_ values of 39.50 and 19.45 μM, respectively. Takahashi et al. reported two new dimeric sesquiterpenoid quinones, nakijiquinones E-F (**92**–**93**), and nine sesquiterpenoid quinones, nakijiquinones J-R (**94**–**102**), from different sponges identified as *Spongia* sp., Sponge SS-1047, SS-265, and SS-1208 [44,45]. Compounds **103**–**106** were isolated from the South China Sea sponge *Dysidea avara* [46]. These compounds possessed the unprecedented “dysidavarane” carbon skeletons. Compound **103** showed growth inhibitory activity against the HeLa cell with an IC_50_ value of 39.9 μM, and dysidavarone D (**106**) showed cytotoxicity against four cell lines (HeLa, A549, MDA231, and QGY7703) with IC_50_ values of 28.8, 21.4, 11.6, and 28.1 μM, respectively. In addition, compounds **103** and **106** also exhibited inhibitory activities against PTP1B with IC_50_ values of 9.98 and 21.6 μM, respectively. Moreover, dysidavarone A (**103**) was totally synthesized in 2013 [47]. In 2012, three new sulfated meroterpenoids (**107**–**109**), including sesquiterpene and hydroquinone moieties, were isolated from the sponge *Aka coralliphaga* collected from southern Mexico [48]. All the compounds showed less potent radical-scavenging effects than Trolox. An indoesesquiterpene hydroquinone (**110**) and a bissesquiterpene hydroquinone (**111**) were isolated from the Thai sponge *Smenospongia* sp. [49]. Additionally, compounds **110** and **111** showed weak cytotoxicity against MOLT-3, HepG2, A549, HuCCA-1, HeLa, HL-60, and MDA-MB-231 with IC_50_ values ranging from 40.0 to 68.2 μM.

An investigation of the marine sponge *Dysidea avara* afforded a new sesquiterpene (-)-N-methylmelemeleone-A (**112**) [50]. Compound **112** showed very weak cytotoxic activity on human colon carcinoma cells, mouse lymphoma cells, and rat hepatoma cells. Compounds **113**–**115** were isolated with unprecedented carbon skeletons from the South China Sea sponge *Dysidea avara* [51]. Dysideanone B (**114**) showed cytotoxicity against the HeLa and HepG2 cell lines with IC_50_ values of 7.1 and 9.4 μM, respectively. Furthermore, compound **114** exhibited significant inhibitory effects on the nitric oxide production induced by lipopolysaccharide at 10 μM. Compounds **116**–**119** were isolated from the Caribbean sponge *Aka coralliphagum* collected off the coast of San Salvador in the Bahamas [52]. Compound **118** exhibited weak activity against the Gram-positive bacteria *Staphylococcus aureus* and *Micrococcus luteus*, and compound **119** showed cytotoxic activities against the L929 mouse fibroblasts, KB-31 epidermoid carcinoma, and breast cancer cell line MCF7. Three new sesquiterpene hydroquinones (**120**–**122**) were isolated from the marine sponge *Dysidea* sp. collected from Okinawa [53]. Compounds **120**–**122** had PTP1B inhibitory activities with IC_50_ values of 11, 9.5, and 6.5 μM, respectively. Two new sesquiterpene aminoquinones (**123**–**124**) were acquired from the sponge *Spongia* sp., which was collected in Vietnam [54]. Compounds **123** and **124** exhibited remarkable antibacterial activities against *Staphylococcus aureus* and *Bacillus subtilis* with MIC values ranging from 6.25 to 12.5 μM. In contrast, they were inactive against *Klebsiella pneumoniae* and *Escherichia coli*. The chemical structures of compounds **90**–**127** are depicted in Figure 7.

Compounds **125**–**128** were isolated from the sponge *Dysidea* sp., which was collected from the South China Sea [55]. Compound **128** showed more potent antibacterial activity against *Escherichia coli*, *Bacillus subtilis*, and *Staphylococcus aureus* than the other compounds. Eight new sesquiterpene quinones/hydroquinones (**129**–**136**) and one solvent-generated artifact (**137**) were isolated from the marine sponge *Spongia pertusa Esper* [56]. Compound **134** displayed CDK-2 affinity with a Kd value of 4.8 μM, and it was the first example of a marine sesquiterpene quinone with CDK-2 affinity. A new sesquiterpene quinone, neoisosmenospongine (**138**), was isolated though methods guided by bioassays. The marine sponge *Dactylospongia metachromia* was collected from Bajotalawaan in North Sulawesi [57]. In 2022, Chen and colleagues isolated four new sesquiterpene hydroquinones, xishaeleganins A–D (**139**–**142**), from the Xisha marine sponge *Dactylospongia elegans* [58]. Compound **140** showed significant antibacterial activity against *Staphylococcus aureus* with a minimum inhibitory concentration value of 1.5 μg/mL. Dysideanones F-G (**143**–**144**) and dysiherbols D–E (**145**–**146**) were isolated from the marine sponge *Dysidea avara* collected from the South China Sea [59]. An anti-inflammatory evaluation showed that dysiherbols D–E exhibited moderate inhibitory activity on TNF-α-induced NF-κB activation in human HEK-293T cells with IC_50_ values of 10.2 and 8.6 μmol·L^–1^, respectively. Li and colleagues isolated six new sesquiterpene quinone/hydroquinone meroterpenoids, arenarialins A–F (**147**–**152**), from the marine sponge *Dysidea arenaria* collected from the South China Sea [60]. Arenarialins A–F showed inhibitory activities on the production of the inflammatory cytokines TNF-α and IL-6 in LPS-induced RAW264.7 macrophages with arenarialin D regulating the NF-κB/MAPK signaling pathway.

Takahashi et al. reported two novel merosesquiterpenoids (**153**–**154**) from different sponges identified as *Spongiidae* sp. [61]. Lastly, compounds **153** and **154** exhibited antimicrobial activities against *Escherichia coli*, *Staphylococcus aureus*, *Bacillus subtilis*, *Micrococcus luteus*, *Aspergillus niger*, *Trichophyton mentagrophytes*, *Candida albicans*, and *Cryptococcus neoformans*. In 2010, a chemical investigation of the sponge *Agelas nakamurai* collected from Menjangan Island in Indonesia afforded two new diterpenes (**155**–**156**), containing the adenine-related moiety [62]. Compound **155** showed antibacterial activity against the planktonic form of *S. epidermidis* (MIC < 0.0877 μM) but did not inhibit biofilm formation. On the contrary, compound **156** showed biofilm formation inhibition but did not inhibit the growth of the planktonic bacteria. Utkina et al. isolated four new sesquiterpenoid arenarone derivatives (**157**–**160**) and a novel dimeric popolohuanone F (**161**) from the CHCl_3_ extract of the Australian marine sponge *Dysidea* sp. [63]. Compound **161** showed 2,2-diphenyl-1-picrylhydrazyl (DPPH) radical scavenging activity with an IC_50_ value of 19 μM. Two new sesquiterpenoids (**162**–**163**) were obtained from a culture broth of the seawater-derived fungus *Trichoderma* sp. TPU1237 isolated from the marine sponge *Dysidea* sp. [64]. The IC_50_ values of compounds **162** and **163** against PTP1B were 53.1 and 65.1 μM, respectively. A few months later, they reported another merosesquiterpenoid (**164**) from another marine sponge *Dysidea* sp. [65]. Its absolute configuration was established by single-crystal X-ray diffraction analysis. Additionally, dysidinoid A (**164**) showed potent antibacterial activity against one strain of pathogenic bacteria, methicillin-resistant *Staphylococcus aureus* (MRSA), with an MIC_90_ value of 8.0 μg/mL. Four new tetracyclic meroterpenes (**165**–**168**) were isolated from an organic extract of the *Dysidea* sp. collected from the Xisha Islands [66]. The chemical structures of compounds **128**–**168** are depicted in Figure 8.

Compound **169** was isolated by bioactivity-guided fractionation from a sponge-derived fungus *Acremonium* sp. obtained from the marine sponge *Stelletta* sp. J05B-1. The unique cyclic skeleton of compound **169** was unprecedented. Compounds **170**–**172** were isolated by bioactivity-guided fractionation from a sponge-derived fungus *Acremonium* sp. obtained from the marine sponge *Stelletta* sp. J05B-1 [67]. Three new nitrogen-containing terpenes (**173**–**175**) were isolated from the sponge *Dysidea robusta* collected in Brazil [68]. Compound **175** was the first furodysinin sesquiterpene derivative with a trans junction between the two six-membered rings of the 1,2,3,4,4a,7,8,8a-octahydro-1,1,6-trimethylnaphthalene moiety. Researchers from the Shanghai Institute of Materia Medica reported two new acetonyl-bearing sesquiterpenes (**176**–**177**) [69]. These metabolites were isolated from the sponge *Dysidea fragilis* collected from Hainan in China. Compounds **176** and **177** showed a weak inhibitory effect with inhibitory rates of 9.9% and 11.5% at a concentration of 20 μg/mL. In a study on the marine sponge *Negombata corticata*, four norterpene-related peroxides (**178**–**181**) were identified from its EtOH extract [70]. Compound **182** and its acetylated derivative (**183**), as well as two new sesquiterpene quinones (**184**–**185**), were isolated from the marine sponge *Dactylospongia elegans*, which was collected from Pugh Shoal, northeast of Truant Island [71]. All the compounds were found to have activities against human tumor cell lines (SF-268, H460, MCF-7, and HT-29) in the range of 1.8–46 μM. In the same year, they obtained three new merosesquiterpenoids (**186**–**188**) that were isolated from the marine sponge *Thorecta reticulata* collected off Hunter Island, Tasmania, Australia [72]. The cytotoxicity of compounds **186**–**188** was also assessed against a panel of human tumor cell lines (SF-268, H460, MCF-7, and HT-29) and a mammalian cell line (CHOK1). All the compounds were found to have 50% growth inhibition activities in the range of 2.1–130 μM, with compound **188** being the most active. Suto and coworkers isolated three new dimeric sesquiterpenoids (**189**–**191**) and one new eudesmane-type sesquiterpenoid (**192**) from the marine sponge *Halichondria* sp. collected from Unten Port in Okinawa [73]. Another six new metabolites of acremines A to R (**193**–**198**) were isolated from the fungus *Acremonium persicinum* cultured from the marine sponge *Anomoianthella rubra* [74]. Kiem and coworkers isolated seven new muurolane-type sesquiterpenes (**199**–**205**) from the marine sponge *Dysidea cinerea* [75].

Researchers from the Tohoku Pharmaceutical University obtained three new unique sesquiterpenes (**206**–**208**) from the marine sponge *Euryspongia* sp. collected from Iriomote Island in Japan [76], and the absolute configuration of **206** was determined after two years [77]. Compounds **209**–**213** were isolated from crude extracts of the marine sponge *Ircinia* sp. [78]. Two terpenoids (**209**, **212**) showed PPARδ agonistic activities with EC_50_ values of 18 and 30 μg/mL, respectively. Wu et al. reported six new caryophyllene-based sesquiterpenoids (**214**–**219**) from the fungus *Hansfordia sinuosae* isolated from the South China Sea sponge *Niphates* sp. [79]. All the compounds showed weak cytotoxic activity against HCT-8, Bel7402, BGC823, A549, and A2780 with IC_50_ values of more than 10 μM. These compounds also showed weak inhibitory effects against the bacterial strains of *Escherichia coli*, *Staphylococcus aureus*, *Bacillus thuringensis*, and *Bacillus subtilis* with MIC values of more than 125 μM. In addition, punctaporonin K (**217**) exhibited potent effects to reduce the triglycerides and total cholesterol at the intracellular level. Compounds **220**–**221** were isolated from a rice solid culture medium of the fungus *Penicillium adametzioides* AS-53 obtained from an undescribed sponge collected from Hainan Island in the South China Sea [80]. Compound **221** showed selective activity against the NCI-H446 cell line with an IC_50_ value of 5.0 μM. An investigation into the Hainan sponge *Axinyssa variabilis* yielded four new uncommon nitrogenous eudesmane-type sesquiterpenes (**222**–**225**) [81]. Compound **226** was isolated from the sponge *Carteriospongia foliascens* collected from the South China Sea [82]. The chemical structures of compounds **169**–**228** are depicted in Figure 9.

Two new structurally unique bridged polycyclic sesquiterpenes (**227**–**228**) and two polycyclic sesquiterpenes (**229**–**230**) were obtained from the marine sponge *Lamellodysidea herbacea*, which was collected in Indonesia [83]. Eight new biosynthetically and chemically related sesquiterpenes (**231**–**238**) were isolated from the southern Australian marine sponge *Dysidea* sp. (CMB-01171) by the GNPS molecular networking approach [84]. Zhang and colleagues isolated ten undescribed eremophilanes, namely, copteremophilanes A-J (**239**–**248**), from a marine sponge *Xestospongia testudinaria*-associated fungus *Penicillium copticola* collected in Weizhou Island [85]. Ohte and colleagues isolated one new unique sesquiterpene lactone, bicyclolamellolactone A (**249**), from the Indonesian marine sponge *Lamellodysidea* sp. [86] The Utkina et al. research group on a different marine sponge *Dysidea* sp. yielded a new unsymmetrical puupehenone-related dimer (**250**) [87]. In addition, diplopuupehenone (**250**) showed DPPH radical scavenging activity with an IC_50_ value of 8 μM. University of Hawaii researchers found a new potent antioxidant and antimicrobial meroterpenoid, puupehenol (**251**), from the deep-water Hawaiian sponge *Dactylospongia* sp. [88]. Compound **251** showed pronounced antioxidant properties and was relatively active towards the Gram-positive bacteria *Staphylococcus aureus* and *Bacillus cereus*. Compound **252** was isolated from the mycelial extract of the marine-derived fungus *Aspergillus aculeatus* CRI323-04 obtained from the sponge *Xestospongia testudinaria*, and it possessed a novel [5,6] fenestrane ring system [89].

Liang and colleagues in Jinan University isolated a pair of new valerenane sesquiterpene enantiomers (**253**–**254**) from the marine sponge *Spongia* sp. [90]. In 2018, researchers from the Shanghai Institute of Materia Medica had reported two rare new furano-trinorsesquiterpenoids (**255**–**256**), which were isolated from the n-BuOH portion of the Beihai sponge *Spongia officinalis* [91]. Further separation led to their corresponding enantiomers. All the new compounds were found to exhibit inhibitory activity against *LasR* and functioned as *Pseudomonas aeruginosa* QS inhibitors. Three new spiro-sesquiterpenoids, myrmekiones A–C (**257**–**259**), were isolated from the marine sponge *Myrmekioderma* sp. collected from the South China Sea [92]. Shen and coworkers conducted a detailed chemical investigation of the South China Sea nudibranch *Hexabranchus sanguineus*, as well as its possible prey the sponge *Acanthella cavernosa*, leading to the isolation of fifteen new nitrogenous sesquiterpenoids, namely, ximaocavernosins A–O, including seven cadinanes (**260**–**266**), seven spiroaxanes (**267**–**273**), and one aromadendrane (**274**) [93]. The same year, they isolated one new aromadendrane-type sesquiterpenoid (**275**) and three new cadinane-type sesquiterpenoids (**276**–**278**) from the Hainan sponge *Acanthella cavernosa*, namely, ximaocavernosin P, (+)-maninsigin D, (+)- and (−)-ximaocavernosin Q [94]. Four new nitrobenzoyl sesquiterpenoids, insulicolides D–G (**279**–**282**), were identified from Antarctican sponge-derived *Aspergillus insulicola* HDN151418. Further studies indicated that insulicolide D (**279**) could significantly suppress cell proliferation to induce apoptosis and blocked the migration and invasion of the pancreatic ductal adenocarcinoma (PDAC) cell lines with IC_50_ values of 2.3–22.9 μM [95,96]. The chemical structures of compounds **229**–**282** are depicted in Figure 10.

Wen and colleagues isolated two new geosmin-type sesquiterpenoids and two new germacrane-type sesquiterpenoids under the guidance of MS/MS-based molecular networking, odoripenoids A–D (**283**–**286**) [97]. These compounds were obtained from the marine mesophotic zone sponge-associated *Streptomyces* sp. NBU3428. Compounds **283**–**284** showed anti-Candida albicans activity with MIC values of 16 and 32 μg/mL, respectively. Hao and colleagues isolated seven new sugar alcohol-conjugated acyclic sesquiterpenes, acremosides A–G (**287**–**293**), from a culture of the sponge-associated fungus *Acremonium* sp. [98]. IMB18-086 was cultivated with the heat-killed *Pseudomonas aeruginosa*. Acremosides A (**287**) and C–E (**289**–**291**) showed significant inhibitory activity against the hepatitis C virus (with EC_50_ values of 4.8–8.8 μM) with no cytotoxicity (CC_50_ of >200 μM). He and colleagues isolated one new marine fungus-derived sesquiterpenoid, Cpd-8 (**294**), that brought about TNF receptor superfamily-induced cell death [99]. Kiem and coworkers isolated an additional five new sesquiterpene derivatives (**295**–**299**) from another marine sponge *Smenospongia cerebriformis* in 2017 [100]. Compounds **295**–**299** showed moderate activity inhibiting LPS-stimulated NO production in BV2 cells with IC_50_ values ranging from 24.37 to 30.43 μM. Compounds **300**–**302** were isolated from the sponge *Verongula* cf. *rigida* collected in Thailand [101]. In particular, compound **300** was the first member of merosesquiterpenes with a polyketide side chain substituted on C-19. One new sesquiterpene, (+)-19-methylaminoavarone (**303**), was isolated from the Xisha Islands marine sponge *Dysidea* sp. It displayed various potent cytotoxic activities [102]. Fourteen undescribed nitrogenous merosesquiterpenoids, purpurols A–D (**304**–**307**) and puraminones A–J (**308**–**317**), were isolated from the sponge *Pseudoceratina purpurea* collected in the South China Sea [103]. In the bioassays, purpurols A and B showed weak anti-inflammation in zebrafish.

The chemical structures of compounds **283**–**317** are depicted in Figure 11, and the remaining information for compounds **66**–**317**, including names and marine sources, is presented in Appendix A.

### 3.2. Diterpenes

Diterpenes are a class of compounds consisting of four isoprene units with a wide range of activities. Compound **318** was isolated from the sponge *Ciocalapata* sp. [104]. It showed antimalarial activity with an IC_50_ value of 0.98 μg/mL. Chanthathamrongsiri et al. reported two new amphilectane-type diterpenes (**319**–**320**) from the sponge *Stylissa* cf. *massa* [105]. Compounds **319** and **320** had two different isonitrile-related functionalities. These two compounds showed antimalarial activity with IC_50_ values of 8.85 and 8.07 μM.

Spongian diterpenes are an important class of diterpenoid products. Compounds **321**–**322** were isolated from an unidentified sponge collected along the coast of Okinawa [106]. They exhibited significant cytotoxicty against NBT-T2 rat bladder epithelial cells with IC_50_ values of 5.6 and 12 μg/mL, respectively. In 2016, four additional new cytotoxic spongian diterpenes (**323**–**326**) were isolated from the acetone extract of *Dysidea* cf. *Arenaria* [107]. The cytotoxicity to NBT-T2 cells was examined, and they displayed IC_50_ values of 3.1, 1.9, 8.4, and 3.1 μM, respectively. Darwinolide (**327**) was isolated from the Antarctic dendroceratid sponge *Dendrilla membranosa* [108]. Darwinolide was cytotoxic against the biofilm phase of the methicillin-resistant *S. aureus* strain and J774 macrophage cell line with IC_50_ values of 33.2 and 73.4 μM. Moreover, it may provide a scaffold for the development of therapeutics for this difficult-to-treat infection. In 2017, an investigation of the sponge *Darwinella* cf. *oxeata* collected in Brazil afforded four new rearranged spongian diterpenoids (**328**–**331**) [109]. A chemical investigation of the Indonesian marine sponge *Spongia ceylonensis* had provided sixteen new spongian diterpene derivatives (**332**–**347**) [110,111,112]. The inhibitory effects on the RANKL-induced osteoclasts in RAW264 macrophages revealed that ceylonin A (**332**) significantly inhibited the formation of multinuclear osteoclasts by 70% in a dose-dependent manner without cytotoxicity, followed by ceylonins E (**336**, 47%), F (**337**, 31%), and D (**338**, 28%).

In 2018, three new spongian diterpenes (**348**–**350**) were isolated from the acetone of the South China Sea sponge *Spongia officinalis* [113]. 3-nor-spongiolide A (**348**) had a rare 3-nor-spongian carbon skeleton, while **349** and **350** had *γ*-butenolide for ring D. A chemical investigation of the marine sponge *Acanthodendrilla* sp. collected in Pulau resulted in the isolation of two new spongian diterpene analogues (**351**–**352**) [114]. Bory and colleagues isolated three new diterpenes, Dendrillins B–D (**353**–**355**), from the Antarctic sponge *Dendrilla antarctica*, which was rich in defensive terpenoids with promising antimicrobial potential [115]. Compound **353** showed single-digit micromolar activity against the leishmaniasis parasite, and **354** had strong hits against MRSA biofilm cultures. Tai and colleagues isolated three new diterpenes, spongenolactones A–C (**356**–**358**), from a Red Sea sponge *Spongia* sp. They were the first 5,5,6,6,5-pentacyclic spongian diterpenes bearing a *β*-hydroxy group at C-1 [116]. All three compounds were found to exert inhibitory activity against superoxide anion generation in fMLF/CB-stimulated human neutrophils. Furthermore, **356** showed a higher activity against the growth of *Staphylococcus aureus*. Dyshlovoy and colleagues isolated a new diterpenen, spongionellol A (**359**), from the sponge *Spongionella* sp. [117]. It exhibited high activity and selectivity in human prostate cancer cells. Jin and colleagues isolated two new diterpenes (**360**–**361**) from the aquaculture sponge *Spongia officinalis*. Compound **360** showed cytotoxicity activity against the K562 cell line with an IC_50_ value of 7.3 μM [118]. Tai et al. isolated four new diterpenes, secodinorspongins A–D (**362**–**365**), from the Red Sea sponge *Spongia* sp. [119]. Compound **362** was found to exhibit inhibitory activity against the growth of *S. aureus*.

Sala and coworkers isolated two nitrogenous rearranged spongian nor-diterpenoids, dendrillic acids A (**366**) and B (**367**), from a marine sponge *Dendrilla* sp. There was mild antiprotozoal activity displayed by dendrillic acid B (**367**) against *Giardia duodenalis* [120]. Two new spongian furanoditerpenes (**368**–**369**) were isolated from a MeOH extract of the sponge *Spongia tubulifera*, collected in the Mexican Caribbean [121]. Compound **368** showed weak biological activity against the A549, A2058, HePG2, MCF-7, and MiaPaca-2 cell lines. Three new furanoditerpenoids (**370**–**372**) were isolated from the CH_2_Cl_2_ extract of the *Spongia* sp. collected from Fiji Islands [122]. One novel C_21_ terpenoidal metabolite (**373**) and two new C_22_ furanoterpenes (**374**–**375**) were isolated from the sponge *Ircinia* sp. collected from Orchid Island, Taiwan [123]. The pharmacological activitiy tests showed that compound **375** exhibited cytotoxicity against the K562, DLD-1, HepG2, and Hep3B cancer cell lines with IC_50_ values of 5.4, 0.03, 0.5, and 1.1 μM, respectively. Furthermore, compounds **374** and **375** were also found to exhibit significant cytotoxicity toward some cell lines. Thus, it is noteworthy to mention that the furan moiety in these compounds was critical for the cytotoxic activity of the C_22_ furanoterpenoids. Further investigation of the sponge *Spongia officinalis* has yielded a C_21_ furanterpene (**376**) [124].

The chemical structures of compounds **318**–**376** are depicted in Figure 12, and the remaining information, including names and marine sources, is presented in Appendix A.

Peter T. Northcote and colleagues isolated eleven new hamigeran-style diterpenes (**377**–**387**) from the New Zealand marine sponge *Hamigera tarangaensis* [125]. Most of these compounds have a unique tricyclic skeleton and moderate levels of cytotoxicity. Later, further investigation of the dictyoceratid marine sponge *Hamigera tarangaensis* yielded seven new members of the hamigeran family of diterpenoids (**388**–**394**) [126]. Most of the new hamigerans exhibited micromolar activity towards the HL-60 promyelocytic leukaemic cell line, and hamigeran G (**378**) also selectively displayed antifungal activity in the budding yeast *Saccharomyces cerevisiae*. Compounds **395**–**403** were isolated from the marine sponge *Hamigera tarangaensis* collected from Cavalli Island, New Zealand [127]. Hamigeran M (**395**) exhibited the strongest degree of potency at a concentration of 6.9 μM, and the other compounds showed cytotoxicity against the HL-60 cell line with IC_50_ values ranging from 14.1 to 33.3 μM. Wojnar and coworkers investigated the extracts of the sponge *Darwinella oxeata*, and they obtained nine new nitrogenous spongian diterpenes (**404**–**412**) [128]. A spectroscopy-guided analysis of a Tongan dictyoceratid marine sponge afforded four new diterpenes (**413**–**416**) [129].

Two new nitrogenous prenylbisabolanes (**417**–**418**) were isolated from the hexane and acetone extracts of the marine Lithistid sponge *Theonella swinhoei*, which was obtained from the coral reef of Iriomote Island in Japan [130]. Compounds **419**–**421** were isolated from an EtOAc extract of the marine sponge *Spongia officinalis* collected from the South China Sea [131]. In addition, all the compounds exhibited moderate inhibition against LPS-induced NO production in RAW264.7 macrophages with IC_50_ values of 12-32 μM. Another chemical study on the marine sponge *Fascaplysinopsis reticulata* has led to the isolation of a new dolabellane-type diterpenoid (**422**) [132]. Luo et al. isolated another new dolabellane diterpene, 6,10,18-triacetoxy-2E,7E-dolabelladiene (**423**), from the South China Sea sponge *Luffariella variabilis* [133]. It demonstrated cytotoxicity towards MCF-7 with an IC_50_ value of 79.2 μM. Wang and colleagues purified seven new kalihinane diterpenoids, kalihioxepanes A-G (**424**–**430**), from the sponge *A. cavernosa* collected in the South China Sea. Kalihioxepane A (**424**) displayed strong cytotoxicity against H69 and K562 tumor cells, while kalihioxepane B (**425**) showed moderate cytotoxicity against K562 cells [134]. Wang and colleagues isolated eight unreported α-acyloxy amides substituted kalihinane diterpenes, named kalihiacyloxyamides A–H (**431**–**438**), from the South China Sea sponge *Acanthella cavernosa* [135]. Compounds **433** and **435** exhibited cytotoxicity against the K562 cell line with IC_50_ values of 6.4 and 6.3 μM, respectively. The IC_50_ values of compounds **435** and **438** against the MDA-MB-231 cell line were 7.3 and 7.9 μM, respectively. The chemical structures of compounds **377**–**438** are depicted in Figure 13.

Thirteen new structurally related meroterpenoids, including three tricyclics (**439**, **446**–**447**), six bicyclics (**440**–**445**), and four monocyclics (**448**–**451**), were isolated from the fungus *Alternaria* sp. JJY-32 obtained from the sponge *Callyspongia* sp. collected off the coast of Hainan Island in China [136]. The NF-κB inhibitory activity of all the compounds showed a weak-to-moderate effect in the RAW264.7 cell line with IC_50_ values ranging from 39 to more than 100 μM. Sixteen new phenylspirodrimane-style sesterterpenes (**452**–**467**) were isolated from the endophytic fungus *Stachybotrys chartarum* collected from the marine sponge *Niphates recondite* [137]. The isoindolone-drimane dimer chartarlactam L (**462**) was determined as a new skeleton. The bioassay results revealed that seven new compounds (**455**–**457**, **462**–**463**, and **465**–**466**) exhibited potent lipid-lowering effects in HepG2 cells in a dose of 10 μM. Additionally, four compounds (**456**–**457**, **462**, and **467**) showed significant inhibition of the intracellular triglyceride (TG) levels in the same cell model, whereas five compounds (**455**–**457**, **464**, and **466**) dramatically reduced total cholesterol (TC).

In 2009, five new diterpene formamides (**468**–**472**) were reported from the tropical marine sponge *Cymbastela hooperi* [138]. In an in vitro antiplasmodial experiment, compound **468** was found to have moderate activity (IC_50_ = 0.5 μg/mL) and compound **469** had weak activity (IC_50_ = 14.8 μg/mL), but **470**–**472** were inactive. Moreover, compound **468** was the rarely natural product that contained both formamide and isonitrile functionalities within the same molecule. Compounds **473**–**476** were isolated from the marine sponge *Spongionella* sp. [139]. All the compounds showed cytotoxic activity against K562 human chronic myelogenous leukemia cells, with IC_50_ values ranging from 4.5 to 15 μM. Meanwhile, they showed slightly toxicity toward the normal PBMC cells, with IC_50_ values ranging from 6.5 to 30 μM. Moreover, all the compounds were proved to be active at a concentration of 100 μM, with gracillin L (**475**) being the most potent (with an inhibition rate = 75%) towards the protein tyrosine kinase EGF-R, and, similarly, as active as the positive control, genistein, which showed an 80% inhibition. Four new spongian-class diterpenes (**477**–**480**) were isolated from the sponge *Dysidea* cf. *arenaria* collected in Okinawa [140]. All the compounds showed cytotoxicity against NBT-T2 rat bladder epithelial cells with IC_50_ values of 10, 1.9, >10, and >10 μg/mL. Four novel 9-N-methyladeninium diterpenoids (**481**–**484**) were isolated from the marine sponge *Agelas* sp. collected in Papua New Guinea [141]. These compounds represented higher unsaturated 9-N-methyladeninium bicyclic diterpenoid derivatives. All the isolated compounds were evaluated for inhibitory activity against *Trypanosoma brucei*, as well as for cytotoxicity against Jurkat cells. Compounds **482** and **483** displayed significant inhibitory action against *T. brucei* with IC_50_ values of 3.6 and 3.0 μg/mL. Moreover, compounds **481**–**483** showed cytotoxicity against Jurkat cells with IC_50_ values of 8.4, 3.3, and 25.0 μg/mL. The chemical structures of compounds **439**–**484** are depicted in Figure 14.

In 2010, a study on the marine sponge *Negombata corticata*, a norterpene-related peroxide (**485**) was also identified from its EtOH extract [70]. Three new unusual C_21_ terpenoids (**486**–**488**) were isolated from a MeOH extract of the marine sponge *Clathria compressa* collected from Panama City in Florida [142]. Clathric acid (**486**) showed an MIC value of 32 μg/mL against *Staphylococcus aureus* (ATTC 6538P) and 64 μg/mL against both methicillin-resistant (ATTC 33591) and vancomycin-resistant *Staphylococcus aureus* (VRSA). One new oxygenated terpenoid (**489**) was isolated from the marine sponge *Coscinoderma matthewsi* collected from the Inner Gneerings Reef, Southeast Queensland, and another two (**490**–**491**) were isolated from another sponge *Dysidea* sp. collected from the same place [143]. A chemical investigation of a MeOH extract of the South China Sea sponge *Cacospongia* sp. yielded two terpenoids belonging to two different skeleton types, including the unusual C_17_ *γ*-lactone norditerpenoids (**492**) and the rare C_21_ pyridine meroterpenoid (**493**) [144]. In 2013, an investigation of the marine sponge *Agelas axifera* led to the isolation of three new pyrimidine diterpenes (**494**–**496**) [145]. All the compounds were found to be moderate inhibitors to cancer cell growth, such as the P388, BXPC-3, MCF-7, SF-268, NCI-H460, KML20L2, and DU-145 cell lines. In addition, axistatins 1–3 (**494**–**496**) had identical antimicrobial profiles, inhibiting Gram-positive bacteria, the exquisitely sensitive Gram-negative pathogen *Neisseria gonorrheae*, and the opportunistic fungus *Cryptococcus neoformans*.

Four new cytotoxic diterpenoid pseudodimers (**497**–**500**) were isolated from the marine sponge *Phorbas gukhulensis* collected off the coast of Gagu-do, Korea [146]. All the compounds exhibited significant cytotoxicity against the K562 and A549 cell lines with IC_50_ values ranging from 0.04 to 0.55 μM. Compound **501** was isolated from the Indonesian sponge *Haliclona* sp. [147]. It showed moderate cytotoxicity against NBT-T2 cells with an IC_50_ value of 4.8 μg/mL and also had antioxidant activity against DPPH with an IC_50_ value of 3.2 μg/mL. Two new IDO inhibitory meroterpenoids (**502**–**503**) were isolated from the sponge *Xestospongia vansoesti* collected in the Philippines [148]. Compounds **504**–**505** were isolated from the fungus *Arthrinium* sp. obtained from the inner tissues of a sponge *Sarcotragus muscarum* collected off the coast of southern Turkey [149]. Compound **506** was isolated from two unidentified species belonging to the genus *Spongia*. In an antibacterial assay, it showed a certain degree of inhibitory effect against *E*. *coli* [150]. It inhibited sea urchin embryo development at a concentration of 20 μg/mL and above, as well as DNA biosynthesis at a dose of 10 μg/mL. A bioassay-guided fractionation of a MeOH extract of the fungus *Arthrinium* sp., isolated from the Mediterranean sponge *Geodia cydonium*, afforded five new diterpenoids (**507**–**511**) [151]. Myrocin D (**511**) inhibited VEGF-A-dependent endothelial cell sprouting with an IC_50_ value of 2.6 μM, whereas the other compounds were inactive. Compounds **512**–**513** were isolated from the fungal strain *Dichotomomyces cejpii* obtained from the *Callyspongia* sp. cf. *C. flammea* [152]. Indole derivative **512** was found to be a CB_2_ antagonist, while compound **513** was identified as the first selective GPR18 antagonist. Compounds **514**–**516** were isolated from an Indonesian sponge of the genus *Spongia* [153]. Compound **516** was unusual, as the D-ring was a pyridyl ring system rather than the standard *δ*-lactone. Compound **831** modestly inhibited aromatase with an IC_50_ value of 34 μM and induced quinone reductase 1 activity with a CD of 11.2 μM, but the remaining isolates were inactive.

In 2015, a new meroditerpene (**517**) was isolated from the EtOH extract of the Okinawan marine sponge *Strongylophora strongilata* [154]. Other researchers from Kumamoto University reported two new additional strongylophorine derivatives (**518**–**519**) isolated from the EtOH extract of another marine sponge *Petrosia corticata* as proteasome inhibitors [155]. An p53–Hdm2 interaction inhibitor, niphateolide A (**520**), was isolated from the Indonesian marine sponge *Niphates olemda* [156]. Compound **517** had PTP1B inhibitory activity with an IC_50_ value of 8.7 μM. Compounds **518** and **519** exhibited potent inhibitory activity against the proteasome with IC_50_ values of 6.6 and 9.3 μM. Then, a new sulfate meroterpenoid named stachybotrin G (**521**) was discovered from the fungus *Stachybotrys chartarum* (MXH-X73) obtained from the marine sponge *Xestospongia testudinaria* [157]. In 2018, three new acetylated terpenoids (**522**–**524**) were isolated from a MeOH extract of the sponge *Rhabdastrella providentiae* collected by scuba divers in Vietnam [158]. Compounds **522**–**524** exhibited potential NO inhibitors with IC_50_ values of 20.4, 17.5, and 46.8 μM, respectively. The chemical structures of compounds **485**–**524** are depicted in Figure 15.

A dimeric C_21_ meroterpenoid (**525**) was isolated from the marine sponge *Dysidea arenaria* [159]. Dysiarenone (**525**) showed inhibitory activity against COX-2 expression and the production of prostaglandin E2 with an IC_50_ value of 6.4 μM in LPS-stimulated RAW264.7 macrophages. A year later, two novel C_19_ terpenoids (**526**–**527**) with an unprecedented carbon skeleton were isolated from an EtOAc extract of the *Stelletta* sp. collected from Vietnamese waters [160]. Compounds **528**–**530** were isolated from the marine sponge *Dysidea septosa* [161]. Septosone A (**528**) featured an unprecedented “septosane” carbon skeleton. Moreover, compound **528** showed in vivo anti-inflammatory activity in CuSO_4_-induced transgenic fluorescent zebrafish likely through the inactivation of the NF-*κ*B signaling pathway. Choi and colleagues isolated one diterpene alkaloid named (–)-agelamide D (**531**) from the marine sponge *Agelas* sp., which could be a natural radiosensitizer in hepatocellular carcinoma models [162].

Kolesnikova and colleagues isolated four new isomalabaricane-derived nor-terpenoids, stellettins S–V (**532**–**535**), from a Vietnamese collection of a *Stelletta* sp. sponge [163]. Among them, compound **532** contained an acetylenic fragment, unprecedented in the isomalabaricane family and extremely rare in other marine sponge terpenoids. Jiang and colleagues isolated three new N-methyladenine-containing diterpenoids (**536**–**538**) from the coralline demosponge *Astrosclera willeyana* collected in Tonga [164]. Four new indole diterpenoids, ascandinines A–D (**539**–**542**), were isolated from an Antarctic sponge-derived fungus *Aspergillus candidus* HDN15-152 [165]. Among them, ascandinine C (**541**) displayed anti-influenza virus A (H1N1) activity with an IC_50_ value of 26 μM, while ascandinine D (**542**) showed cytotoxicity against HL-60 cells with an IC_50_ value of 7.8 mM. A chemical investigation of a Red Sea *Spongia* sp. led to the isolation of 17-dehydroxysponalactone (**543**), which was found to significantly reduce superoxide anion generation and elastase release at a concentration of 10 μM [166]. Zou and coworkers isolated two new meroterpenoids, guignardones Y–Z (**544**–**545**), from the fungus *Penicillium* sp. NBUF154, which was obtained from a 60m deep *Crella* sponge [167]. A biological evaluation showed that compound **544** exerted a potent inhibitory effect towards human EV71. Pech-Puch and coworkers isolated three new diterpene alkaloids (**546**–**548**) from the sponge *Agelas citrina*, collected on the coasts of the Yucatán Peninsula [168]. An evaluation of the antimicrobial activity against the Gram-positive pathogens showed that all of them were active, with (+)-10-epiagelasine B (**547**) being the most active compound with an MIC value in the range of 1–8 μg/mL. Cho and colleagues isolated three norditerpene cyclic peroxides (**549**–**551**) from the marine sponge *Diacarnus spinipoculum* [169]. Prieto et al. isolated one previously unreported 9,11-dihydrogracillinone A (**552**) from the sponge *Dendrilla antarctica*. The results obtained from experiments clearly indicated a potent antifouling activity for compound **552** [170]. Shen and colleagues isolated six new diterpenoids (**553**–**558**) from the sponge *Chelonaplysilla* sp. [171]. An unprecedented monocyclic diterpenoid featuring a 2,7-ring-opened halimane-type skeleton, echinohalimane B (**559**), and a new subersin-type diterpenoid, oculatolide B (**560**), were isolated from the sponge *Sarcotragus* sp. [172]. Wang and colleagues isolated a novel rearranged pimarane diterpenoid, pestanoid A (561), from the *Chalinidae* sp. sponge-derived fungus *Pestalotiopsis* sp. NBUF145 [173]. Compound **561** inhibited bone marrow monocyte osteoclastogenesis in vitro with IC_50_ values of 4.2 ± 0.2 μM.

The chemical structures of compounds **525**–**561** are depicted in Figure 16, and the remaining information for compounds **377**–**561**, including names and marine sources, is presented in Appendix A.

### 3.3. Sesterterpenes

Sesterterpenes are rare in terrestrial plants and are mainly found in sponges and their associated microorganisms, such as coscinolactams A and B (**562**–**563**), two novel nitrogen-containing cheilanthane sesterterpenoids, isolated from the marine sponge *Coscinoderma mathewsi* [174]. In addition, these two compounds showed moderate anti-inflammatory activity, measured as their capability to inhibit PGE2 and NO production. Sesterterpenes have three main structural types: manoalide-type, hyrtiosane-type, and scalarane-type. Of these, the scalarane-type is the most common [175]. A bioassay-guided separation led to the isolation of three new scalarane sesterterpenes (**564**–**566**) from the acetone extract of the South China Sea sponge *Phyllospongia foliascens* [176]. Moreover, compound **565** showed cytotoxic activity against the P388 leukemia cell line with an IC_50_ value of 6.5 μg/mL. One year later, the same authors reported an additional new scalarane sesterterpene (**567**) from the same sponge [177]. A chemical investigation of an EtOAc-soluble fraction of the methanolic extract of the Thai sponge *Hyrtios gumminae* led to the isolation of four new sesterterpenes (**568**–**571**) [178]. Compounds **572**–**575** were isolated from the Indonesian marine sponge *Carteriospongia foliascens*, which was harvested near Makassar, Indonesia [179]. Compound **572** had moderate inhibition of RCE-Protease with an IC_50_ value of 38 μg/mL. Furthermore, the cytotoxic effect of **572** against four cell lines (PC-3, LoVo, CACO-2, and MDA-468) was more active than **573** with IC_50_ values of less than 10.0 μg/mL. In a parallel study on the marine sponge *Carteriospongia foliascens*, 15 novel sesterterpenes (**576**–**590**) were identified from its organic extract [82]. In addition, compound **579** showed remarkable antifouling activity against barnacle *Balanus amphitrite* and lethal activity against *A. salina* with EC_50_ values less than 2.5 μg/mL, LD_50_ values between 4.0 and 10.0 μg/mL. Compounds **591**–**595** were isolated from the sponge *Hyatella* sp., collected off the coast of Soheuksan-do in Korea [180]. These compounds exhibited moderate cytotoxicity and antibacterial activity and weak inhibitory activity against isocitrate lyase.

Three novel scalarane sesterterpenes (**596**–**598**) were isolated from a Korean marine sponge *Psammocinia* sp. Furthermore, they exhibited cytotoxicity against the intractable human cancer cell lines A498, ACHN, MIA-paca, and PANC-1, with IC_50_ values ranging from 0.4 to 48 μM [181]. In 2014, five new scalarane derivatives (**599**–**603**) were isolated from the organic extract of two Thorectidae sponge samples, *Hyrtios* sp. and *Petrosaspongia* sp., collected from the Fiji Islands [182]. Furthermore, these molecules were tested by AlphaScreen at several concentrations from 50 pM to 50 μM to deeply explore their inhibition profile. The results showed activities for compounds **600** and **601** with IC_50_ values of 0.6 μM and 0.4 nM. Compound **600** was also the most potent compound in reducing the binding of TDP-43 to its cognate DNA. Compounds **604**–**605** were isolated from a MeOH extract of the South China marine sponge *Hyrtios erectus* [183]. In 2015, a biology-guided fractionation of the Cl_2_H_2_ fraction of the marine sponge *Phyllospongia lamellose* collected from Shaab Saad, 13 km north of Hurghada, along the Red Sea, yielded five new scalarane-type sesterterpenes (**606**–**610**) [184]. All the compounds had different inhibitory activities against three cancer cell lines (MCF-7, HCT-116, and HePG-2) at the highest concentration of 10 μg/mL when compared to doxorubicin. Compound **610** showed the strongest inhibitory activity against Gram-positive strains. Peng and colleagues isolated six new 24-homoscalaranes, lendenfeldaranes E-J (**611**–**616**), from the marine sponge *Lendenfeldia* sp. [185]. Compounds **611**–**613** were proven to be the first anti-neutrophilic scalaranes.

The chemical structures of compounds **562**–**616** are depicted in Figure 17, and the remaining information, including names and marine sources, is presented in Appendix A.

In 2021, Shin and colleagues identified 15 new scalarane-type sesterterpenoids (**617**–**631**) from the marine sponge *Dysidea* sp., which was collected from the Province of Bohol in the Philippines [186]. The biological properties of all the compounds were evaluated using the MDA-MB-231 cancer cell line. Compound **623**, which bears a pentenone E-ring, exhibited significant cytotoxicity with a GI_50_ value of 4.21 μM. Dysiscalarones A–E (**632**–**636**) were isolated from the marine sponge *Dysidea granulosa* collected from the South China Sea [187]. Dysiscalarones A–B (**633**–**634**) showed NO production inhibitory activity with respective IC_50_ values of 16.4 and 18.5 μM. Peng and colleagues isolated seven new homoscalaranes (**637**–**643**) from the marine sponge *Lendenfeldia* sp. collected in Taiwan [188]. Among them, compounds **637**–**639** exhibited potential anti-inflammatory activity. Chakraborty and Francis isolated hyrtioscalaranes A and B (**644**–**645**) from the Demosponge *Hyrtios erectus* [189]. They both exhibited significant anti-inflammatory activity and antioxidant activity. Tran and colleagues isolated eight new scalarane sesterterpenoids (**646**–**653**) from the sponge *Hyrtios erectus* [190]. All of them were found to show weak growth inhibitory activity against HeLa and MCF-7 cells, with a minimal IC_50_ value of 20.0 μM. Yu and colleagues isolated eight new scalarane sesterterpenes, phyllofenones F-M (**654**–**661**), from the marine sponge *Phyllospongia foliascens* collected from the South China Sea [191]. Among them, compounds **657** and **659** displayed weak inhibitory activity against *S. aureus* and *E. coli*, with MIC values of 16 and 8 μg/mL, respectively. Compounds **654**–**661** exhibited cytotoxic activity against the HeLa, HCT-116, H460, and SW1990 cancer cell lines, with IC_50_ values ranging from 3.4 to 19.8 μM. Yu and colleagues isolated five new scalarane derivatives featuring an unprecedented 6/6/6/5 tetracyclic dinorscalarane scaffold, phyllospongianes A-E (**662**–**666**), from the marine sponge *Phyllospongia foliascens* [192]. Compounds **662**, **663**, and **664** exhibited antibacterial activity against *V. vulnificus*, *V. parahemolyticus*, *E. coli*, *S. aureus*, *E. faecalis*, *B. subtilis*, and *P. aeruginosa* with MIC values ranging from 1 to 8 μg/mL. Furthermore, compound **661** exhibited significant cytotoxic activity on the MDA-MB-231, HepG2, C4-2-ENZ, MCF-7, H460, and HT-29 cancer cell lines with IC_50_ values in a range between 0.7 and 13.2 μM. Compound **667** and one new scalarane sesterterpenoid (**668**) were isolated from the sponge *Hippospongia* sp. collected from coral reefs in Taiwan [193]. In 2020, nine new C_27_ bishomoscalarane sesterterpenes (**669**–**677**) and five new C_26_ 20,24-bishomo-25-norscalarane sesterterpenes (**678**–**682**) were isolated from a MeOH extract of the sponge *Dysidea granulosa* collected from the Xisha Islands of the South China Sea [194]. Compound **672** exhibited moderate antiproliferative activity against the HCT116, A-649, BEL-7402, and Jurkat cell lines with GI_50_ values of 6.4, 8.1, 11, and 13 μM, respectively. A new sesterterpene (**683**) was isolated from the acetone extract of the sponge *Dysidea* sp. in 2010 [195]. The chemical structures of compounds **617**–**683** are depicted in Figure 18.

Compounds **684**–**686** were obtained from the New Zealand marine sponge *Semitaspongia bactriana* with toxicity against the diatom *Nitzschia closterium* and bryozoan *Bugula neritina* [196]. Furthermore, all three compounds exhibited toxicity against *B. neritina* with EC_50_ values of 7.41, 1.22, and 1.59 μM, respectively. Studies on the organic extract of the marine sponge *Hamigera* sp. offered two new sesterterpenoids (**687**–**688**) [197]. Alotaketals A (**687**) and B (**688**) have an unprecedented alotane carbon skeleton, and they significantly activate the cAMP cell signaling pathway with EC_50_ values of 18 and 240 nM, respectively. One year later, ansellone A (**689**) was isolated from the sponge *Phorbas* sp. [198]. It was a new sesterterpenoid with the unprecedented “ansellane” carbon skeleton, and it also activated the cAMP signaling in HEK293 cells in the absence of hormone binding with an EC_50_ value of 14 μM. Later, four new sesterterpenoids (**690**–**693**) were isolated from specimens of the sponge *Phorbas* sp. collected in British Columbia [199]. Alotaketal C (**693**) activated cAMP signaling in HEK293 cells with an EC_50_ value of 6.5 μM. Compounds **694**–**696**, with a spiroketal of the hydrobenzoyran moiety, were isolated from the Korean marine sponge *Phorbas* sp. [200]. Phorbaketal A (**694**) exhibited mild activity against human colorectal cancer HT-29 with an IC_50_ value of 12 μg/mL, hepatoma cancer HepG2 with an IC_50_ value of 11.2 μg/mL, and lung cancer A549 with an IC_50_ value of 11 μg/mL. Further investigation of the marine sponge of the genus *Phorbas* collected from Gageo Island, Korea, reported three additional sesterterpenoids named phorbaketals L–N (**697**–**699**) [201]. The cytotoxicity of compounds **697**–**699** was evaluated against three human cancer cell lines using the MTT assay. The results showed that compound **699** exhibited potent activity against the human pancreatic cancer cell line Panc-1 and the renal cancer cell lines A498 and ACHN with IC_50_ values of 11.4 μM, 18.7, and 24.4 μM, respectively. An additional nine new sesterterpenoids (**700**–**708**) were isolated from another Korean sponge *Monanchora* sp. [202]. The absolute configurations of these compounds were defined using the modified Mosher’s method and a CD spectroscopic data analysis. Compounds **704** and **705** showed weak cytotoxicity against the A498 human renal cancer cell line. Compound **708** showed moderate activity against all four human cancer cell lines, while the others were inactive. University of Auckland researchers achieved a total synthesis of **708** in 2015 [203]. Later, two new sesterterpenoids (**709**–**710**) were isolated from the Korean marine sponge *Phorbas* sp. [204]. The absolute stereochemistry of compound **709** was determined by the Mosher ester method and exhibited a positive effect on the calcium deposition activity in C_3_H_10_T_1/2_ cells. The same group at Seoul National University reported two unprecedented sesterterpenoids, phorone A (**711**) and isophorbasone A (**712**), which were isolated from Korean marine sponge *Phorbas* sp. [205]. The chemical structures of compounds **684**–**712** are depicted in Figure 19.

The chemical investigation of a new collection of the sponge *Ircinia formosana* was carried out, which resulted in the isolation of seven new linear C_22_-furanosesterterpenoids (**713**–**719**) [206]. Among these compounds, compound **717** exhibited significant inhibition of the peripheral blood mononuclear cell proliferation induced by phytohemaglutinin. In 2011, further investigation of the sponge *Spongia officinalis* yielded a linear furanosesterterpene (**720**) [124]. A chemical investigation of the marine sponge *Ircinia oros* yielded two linear furanosesterterpenoids (**721**–**722**). They were the first examples of rare glycinyl lactam-type sesterterpenes [207]. In addition, these two compounds appeared to be moderately active against all protozoan parasite (*P. falciparum*, *T. brucei rhodesiense*, *T. cruzi*, and *L. donovani*) activities with IC_50_ values between 28 and 130 μM. Three new furanosesterterpene tetronic acids (**723**–**725**) were isolated from the *Psammocinia* sp., which was collected in North Sulawesi, Indonesia [208]. Sulawesins A and B (**723**–**724**) possessed unprecedented 5-(furan-3-yl)-4-hydroxycyclopent-2-enone moiety. Sulawesin C (**725**) was found to be the first dimer in this family. The inhibitory effects of compounds **723** and **724** towards USP7 had IC_50_ values of 2.9 and 4.6 μM, respectively. The absolute configuration of two new peroxiterpenes (**726**–**727**) was determined by the modified Mosher’s method [209]. These two compounds were isolated from the marine sponge *Diacarnus bismarckensis*. Compound **726** showed potential bioactivity against *T. brucei* with an IC_50_ value of 2 μg/mL. Two new cyclic peroxide norsesterterpene derivatives (**728**–**729**) were isolated and characterized from the acetone extract of the Red Sea sponge *Diacarnus erythraeanus* [210]. These two compounds displayed mean IC_50_ growth inhibitory activity against the Hs683, U373, A549, MCF-7, PC-3, LoVo, and B16F10 cell lines at concentrations of less than 10 μM. A chemical investigation of the sponge *Diacarnus megaspinorhabdosa* resulted in the isolation of one new norsesterpene cyclic peroxide, megaspinoxide A (**730**) [211]. Compound **730** showed strong activity against *Bacillus cereus*, *Staphylococcus aureus*, and *Candida albicans* at a concentration of 100 μg/disc. Eight new acyclic manoalide-related sesterterpenes (**731**–**738**) were isolated from the South China Sea sponge *Hippospongia lachne* [212]. The absolute configurations were determined by the modified Mosher’s method and CD data. Compound **731** exhibited cytotoxicity against the A549, HeLa, and HCT-116 cell lines with IC_50_ values of 0.05, 0.048, and 9.78 μM, respectively. Compound **731** also showed moderate PTP1B inhibitory activity with an IC_50_ value of 23.81 μM, and compound **732** also showed moderate cytotoxicity against the HCT-116 cell line and PTP1B inhibitory activity with IC_50_ values of 35.13 and 39.67 μM, respectively. In addition, compounds **732** and **735** showed weak anti-inflammatory activity, with IC_50_ values of 61.97 and 40.35 μM for PKC*γ* and PKC*α*, respectively. Compounds **739**–**746** were isolated from the sponge *Coscinoderma* sp., collected from Chuuk Island in Micronesia [213]. These compounds exhibited moderate cytotoxicity against the K562 cell line and inhibitory activity against isocitrate lyase and sortase A, as well as Na^+^/K^+^-ATPase. Compounds **747**–**748** were isolated from the fungus *Aspergillus insuetus* isolated from the Mediterranean sponge *Petrosia ficiformis* [214]. Compounds **747** and **748** were found to be inhibitors of the integrated chain (NADH oxidase activity) with IC_50_ values of 3.90 and 2.97 μM, respectively.

Robert J. Capon and colleagues from the University of Queensland found a new meroterpene sulfate fascioquinol A (**749**) together with two new meroterpenes, fascioquinol E (**750**) and fascioquinol F (**751**), from the *Fasciospongia* sp., which was collected from southern Australian deep waters [215]. These compounds displayed little or no inhibitory activity towards human cell lines. However, compound **749** displayed potential Gram-positive selective antibacterial activity towards *S. aureus* and *B. subtilis*. Compounds **752**–**756** were obtained from the fungus *Aspergillus* sp., which was isolated from the sponge *Tethya aurantium* collected in Italy [216]. These compounds were found to be similar with the known austalides A-L previously isolated from *Aspergillus ustus*. Later, an additional three new austalides (**757**–**759**) were isolated from a sponge-derived fungus *Aspergillus aureolatus* HDN14-107 [217]. Compound **759** exhibited weak activity against H1N1 with an IC_50_ value of 90 μM. Two new sesterterpenoids (**760**–**761**) were isolated from the marine sponge *Cateriospongia flabellifera*, which was collected in Vanuatu [218]. These two compounds had moderate growth inhibitory activity with IC_50_ values around 10–20 μM. In 2013, a report on the marine sponge *Hyrtios communis* yielded six novel sesterterpene analogues (**762**–**767**) [219]. Among these compounds, three of them (**762**–**766**) were the potent inhibitors of hypoxia (1% O_2_)-induced HIF-1 activation with IC_50_ values of 3.2, 3.5, and 6.2 μM, respectively.

The chemical structures of compounds **713**–**767** are depicted in Figure 20.

Compounds **768**–**772** were isolated from the EtOH extract of the sponge *Hippospongia lachne* collected off Yongxing Island in the South China Sea [220]. Compounds **768** and **772** showed moderate PTP1B inhibitory activity with IC_50_ values of 5.2 and 8.7 μM, respectively. Compounds **770** and **771** exhibited weak PTP1B inhibitory activity with IC_50_ values of 33 and 14 μM, respectively. They also evaluated the cytotoxicity of these compounds against the A549, HeLa, and HCT-116 cancer cell lines, and only compound **768** exhibited weak activity against the HCT-116 cell line with an IC_50_ value of 11.6 μM. Woo and colleagues reported three tetracyclic sesterterpenes (**773**–**775**) from the Korean marine sponge *Clathria gombawuiensis*, which was collected from Chuuk Island, Micronesia [221]. All the compounds showed cytotoxicity against two cancer cell lines (K562 and A549) with IC_50_ values around 0.77–4.65 μM when compared with doxorubicin (IC_50_ = 0.79, 0.70 μM). Furthermore, compounds **773** and **775** showed moderate antibacterial activity, while their diastereomer **774** was virtually inactive. The same trend was also observed for the inhibition of the enzymes Na^+^/K^+^-ATPase and isocitrate lyase (ICL), which can be attributed to the 3-dimensional structure of spiroketal. Compound **776** was isolated from an EtOAc extract of the undescribed marine sponge-associated fungus *Aspergillus similanensis* KUFA 0013 [222]. It was evaluated for antimicrobial activity against Gram-positive and Gram-negative bacteria, *Candida albicans* ATCC 10231, and multidrug-resistant isolates from the environment. Chevalone E (**776**) showed synergism with the antibiotic oxacillin against MRSA. Two new unique sesterterpenes (**777**–**778**) with PTP1B inhibitory activity were isolated from the Indonesian marine sponge *Hyattella* sp. [223]. The inhibitory activity was 7.45 μM for compounds **777**, and compound **778** showed a 42% inhibition at a concentration of 24.2 μM. Compounds **779**–**780** were isolated from the marine sponge *Clathria gombawuiensis* collected from Korean waters [224].

In 2016, eight new sesterterpenoids (**781**–**788**) were isolated from a MeOH extract of the marine sponge *Phorbas* sp. collected in Howe Sound, British Columbia [225]. Compounds **781** and **787** were found to induce HIV proviral gene expression. In the next year, a pair of enantiomeric sesterterpenoids (**789**) were isolated from the EtOH extract of the marine sponge *Hippospongia lachne* collected from South China Sea [226]. Compounds **789a** and **789b** showed potent antifungal activity against three strains of hospital-acquired pathogenic fungi, *Candida albicans* SC5314, *Candida glabrata* 537, and *Trichophyton rubrum*, with MIC_50_ values between 0.125 and 0.25 μg/mL. A chemical investigation of the Mediterranean Homoscleromorpha sponge *Oscarella balibaloi* isolated and identified a new family of simple glucosylated sesterterpenes (**790**–**793**), named balibalosides [227]. In a study of the sponge *Cacospongia* sp., additional skeleton types, C_25_ manoalide-type sesterterpenoids (**794**–**796**), were isolated [144]. Oshimalides A (**797**) and B (**798**) were isolated from the *Luffariella* sp. marine sponge [228]. Luo and coworkers isolated eleven rare acyclic manoalide derivatives (**799**–**810**) from the sponge *Luffariella variabilis* collected in the South China Sea. Compounds **799**–**805** and **809** demonstrated cytotoxic activity against several human cancer cell lines with IC_50_ values ranging from 2 to 10 μM [229]. Yu and colleagues isolated five new *γ*-oxygenated butenolide sesterterpene derivatives, dactylospenes A–E (**811**–**815**). Compounds **811** and **813** exhibited moderate cytotoxicity against the DU145, SW1990, Huh7, and PANC-1 cancer cell lines with IC_50_ values in the range of 2.11–13.35 μM [230]. The chemical structures of compounds **768**–**815** are depicted in Figure 21.

Xu and colleagues, in research on the sponge *Sarcotragus* sp., found four new butenolide sesterterpenes featuring a rare methyl-transferred 6/6/6-tricyclic fused ring system with a butyrolactone moiety, Sarcotragusolides A–D (**816**–**819**); a *γ*-hydroxybutenolide sesterterpene derivative (**820**); and a new scalarane sesterterpene (**821**). Compounds **816a**, **816b**, and **817** presented modest cytotoxic activity against several human cancer cell lines [172]. Hang and colleagues isolated four new sesterterpenes, hippotulosas A–D (**822**–**825**), from the marine sponge *Hippospongia fistulosa* 1889 [231]. Bracegirdle and colleagues found several new suberitenone derivatives and terpenoids from the Antarctic sponge *Suberites* sp.: neosuberitenone (**826**), with a new carbon scaffold herein termed the ‘neosuberitane’ backbone; six suberitenone derivatives (**827**–**832**); an ansellane-type terpenoid (**833**); and a highly degraded sesterterpene (**834**). Suberitenone F (**831**) was found to be active against RSV in the biological activity test [232].

Three new natural marine compounds, isosuberitenone B (**835**), 19-episuberitenone B (**836**), and isooxaspirosuberitenone (**837**), were isolated from the Antarctic sponge *Phorbas areolatus* [233]. Compounds **835**–**837** represented new chemical entities of suberitane sesterterpenoids. These compounds were found to exhibit moderate cytotoxic activity, whereas **835** and **836** displayed significant activity against the A549, HepG2, HT-29, and MCF-7 cell lines with IC_50_ values of less than 10 μM. Majer and coworkers isolated two new ircinianin-type sesterterpenoids, ircinianin lactone B (**838**) and ircinianin lactone C (**839**), from the marine sponge *Ircinia wistarii*. Ircinianin lactones B and C represented new ircinianin terpenoids with a modified oxidation pattern [234]. Compounds **840**–**842** were isolated from an organic extract of the marine sponge *Fasciospongia* sp. collected in Palau [235]. These compounds exhibited inhibitory activity against *Streptomyces* 85E in the hyphae formation inhibition assay. Only compound **842** demonstrated a moderate cytotoxic effect on MCF-7 (IC_50_ = 13.4 μM), LNCaP (IC_50_ = 21.8 μM), and LU-1 cells (IC_50_ = 5.0 μM), respectively. A study on the marine sponge *Negombata corticata*, a norterpene-related peroxide (**843**), was identified from its EtOH extract [70]. Compounds **844**–**846** were obtained from the EtOAc extract of the marine-derived fungus *Aspergillus insuetus*, which was isolated from the Mediterranean sponge *Psammocinia* sp. [21]. Compound **844** exhibited weak anti-fungal activity towards *N. crassa* with an MIC value of 140 μM. Meanwhile, compound **846** showed mild cytotoxicity towards MOLT-4 human leukemia cells. The chemical structures of compounds **816**–**846** are depicted in Figure 22.

Seven new phenylspirodrimanes (**847**–**853**) were isolated from the fungus *Stachybotrys chartarum* MXH-X73, which was obtained from the sponge *Xestospongia testudinaris* collected from Xisha Island, China [236]. All the compounds were tested for their antiviral activity against wild-type HIV-1 replication. Only **847** exhibited an inhibitory effect on HIV-1 replication with an EC_50_ value of 8.4 μM at a final concentration of 10 μM. A further study showed that **847** could block NNRTI-resistant strains (HIV-1RT-K103N, HIV-1RT-L100I, K103N, HIV-1RT-K103N, V108I, HIV-1RT-K103N, G190A, HIV-1RT-K103N, and P225H) and wild-type HIV-1 (HIV-1wt) with EC_50_ values of 7.0, 23.8, 13.3, 14.2, 6.2, and 8.4 μM, respectively. Shin and colleagues from Seoul National University reported five suvanine-lactam derivatives (**854**–**858**) from the Korean marine sponge *Coscinoderma* sp., which was collected from Chuuk Island, Micronesia [237]. All the compounds showed cytotoxicity against two cancer cell lines (K562 and A549). Researchers from Peking University reported six new DMOA-related meroterpenoids (**859**–**864**) from an EtOAc extract of the unidentified sponge-associated fungus *Penicillium brasilianum* [238]. Additionally, biological tests revealed that only **859** significantly stimulated the expression of filaggrin and caspase-14 in HaCaT cells in a dose-dependent manner, while compounds **860** and **861** displayed moderate inhibition against NO production in LPS-induced RAW 264.7 macrophages. It was indicated that **859** could reduce UVB-induced cell damage. In addition, compounds **861**–**864** also inhibited the DNA expression of the HBV virus in HepG2.2.15 cells with inhibitory rates of 25, 15, and 10%, respectively. One year later, six additional bioinformatics (**865**–**870**) were isolated from the same fungus [239]. Brasilianoid L (**870**) exhibited significant inhibition against bacteria invasion into host cells. An investigation of the sponge *Dysidea villosa* collected from the South China Sea resulted in the isolation of four unusual merosesterterpenoids (**871**–**874**) [240]. Dysivillosins A-D are the first natural products of terpenepolyketide–pyridine hybrid metabolites. An anti-allergic activity evaluation showed that compounds **871**–**874** potently inhibited the release of β-hexosaminidase with IC_50_ values ranging from 8.2 to 19.9 μM. Additionally, these four meroterpenoids could downregulate the production of lipid mediator leukotrienes B4 (LTB4) and the pro-inflammatory cytokine interleukin-4 (IL-4) in the antigen-stimulated RBL-2H3 mast cells. Further biological investigations exhibited that dysivillosin A (**871**) could suppress the phosphorylation of Syk and PLC*γ*1 in the IgE/FcɛRI/Syk signaling pathway.

A diketopiperazine-indole alkaloid that was isolated from the marine sponge *Ircinia variabilis*-derived fungus *Eurotium* sp. was named fintiamin (**875**) [241]. Fintiamin was composed of amino acid and terpenoid moieties resulting in a terpenoid–dipeptide derivative, which shows affinity for the cannabinoid CB_1_ receptor. Reiko Tanaka and colleagues from the Osaka University of Pharmaceutical Sciences reported six decalin derivatives (**876**–**881**) isolated from the strain of *Trichoderma harzianum* OUPS-111D-4 originally derived from the marine sponge *Halichondria okadai* [242,243,244,245]. In addition, compounds **879**–**880** exhibited the most inhibitory activity against the P388, HL-60, and LI210 cell lines with IC_50_ values of less than 10 μM, followed by **876**–**878** with IC_50_ values of 54.5, 42.2, and 41.3 μM, respectively. A chemical investigation of the Red Sea *Spongia* sp. led to the isolation of a new furanyl trinorsesterpenoid, 16-*epi*-irciformonin G (**882**) [166]. Two new meroterpenoids, hyrtamide A (**883**) and hyrfarnediol A (**884**), were isolated from the South China Sea sponge Hyrtios sp. in Wang’s group [246]. They found that compound **883** exhibited weak cytotoxicity against HCT-116 with an IC_50_ value of 41.6 μM. The sponge *Ircinia felix* was selected for the significant anti-human adenovirus (HAdV) activity displayed by its organic extracts [247]. Its chemical analysis yielded three novel sesterterpene lactams, ircinialactams J-L (**885**–**887**). Ircinialactam J displayed significant antiviral activity against HAdV without significant cytotoxicity, showing an effectiveness 11 times greater than that of the standard treatment, cidofovir^®^.

The chemical structures of compounds **847**–**887** are depicted in Figure 23, and the remaining information for compounds **617**–**887**, including names and marine sources, is presented in Appendix A.

### 3.4. Triterpenes

Common triterpenoids belong to tetra- and/or pentacyclic compounds, and some are usually combined with one or several carbohydrate chains. The isomalabaricane triterpenoids, constructed of tricyclic core systems with long side chains, represent those with a marine origin [244]. In 2009, nine new terpenoids (**888**–**896**) were obtained from the Red Sea sponge *Callyspongia* (=*Siphonochalina*) *siphonella* [248]. The identification of the isolated compounds was achieved using X-ray crystallography and extensive spectral analyses. Additionally, all the compounds were evaluated for their ability to reverse P-glycoprotein (P-gp)-mediated multidrug resistance in human epidermoid cancer cells. At a noncytotoxic concentration of 10 μM, sipholenone E (**889**) displayed better activity than a positive control in reversing P-gp-mediated MDR to colchicine. Sipholenol L (**893**) and siphonellinol D (**895**) also showed P-gp modulatory activity. A year later, seven new isomalabaricane derivatives (**897**–**903**) and a new monocyclic triterpene glycoside (**904**) were isolated from a MeOH extract of *Rhabdastrella globostellata* collected from Amamioshima, Japan [249]. Compounds **900**–**903**, possessing a cyclopentane side chain, exhibited weak activity against the proliferation of promyelocytic leukemia HL-60 cells with IC_50_ values of 21, 29, 44, and 11 μM.

Five unique isomalabaricane-type triterpenoids (**905**–**909**) were isolated from the Okinawan marine sponge *Rhabdastrella* cf. *globostellata* [250]. In 2012, a chemical investigation of the marine sponge *Rhabdastrella globostellata* yielded nine unprecedented isomalabaricane-type triterpenoids (**910**–**918**) [251]. Globostelletins J–R (**910**–**918**) were characterized as an unusual cyclopentane unit linked at different positions of their side chains. Compounds **910** and **911** showed more potent inhibitory effects against ALK, FAK, IGF-1R, SRC, and VEGF-R2 human tumor-related protein kinases than the other compounds. These findings indicated that these two compounds may be developed as protein kinase-targeted inhibitors. A new isomalabaricane triterpene (**919**) was isolated from the crude Et_2_O-soluble extract of the Hainan sponge *Stelletta* sp. [252]. A chemical investigation of the marine sponge *Jaspis stellifera* yielded seven new isomalabaricane-type triterpenoid derivatives (**920**–**926**) [253,254]. In 2021, Kolesnikova and colleagues isolated two new isomalabaricane triterpenoids, stellettins Q and R (**927** and **928**), from a Vietnamese collection of the *Stelletta* sp. sponge [163]. Lai isolated two new isomalabaricane-type triterpenes, rhabdastin H (**929**) and rhabdastin I (**930**), from the marine sponge *Rhabdastrella* collected in Kenting, Taiwan [255]. Both compounds **929** and **930** showed activity against K562 (with IC_50_ values of 11.7 and 9.8 μM) and Molt4 (with IC_50_ values of 16.5 and 11.0 μM) leukemic cells in an MTT cell proliferation assay.

The chemical structures of compounds **888**–**930** are depicted in Figure 24, and the remaining information, including names and marine sources, is presented in Appendix A.

In 2022, Chen and colleagues isolated nine new isomalabaricane terpenoids (**931**–**939**) from the sponge *zza globostellata* from Ximao Island [256]. Seven new triterpene glycosides (**940**–**946**) were isolated from the sponges *Erylus formosus* and *Penares* spp. [257]. The chemical structures of compounds **931**–**946** are depicted in Figure 25.

Five new nortriterpene glycosides (**947**–**951**) were isolated from the tropical sponge *Lipastrotethya* sp. collected from Chuuk in Micronesia [258]. Several compounds exhibited cytotoxicity against the A549 and K562 cell lines, as well as weak inhibitory activity against Na^+^/K^+^-ATPase. The first C_3_-pyranosyl 4-hydroxypyrone structure (**952**) was isolated from the fungus *Epicoccum* sp. JJY40, obtained from the sponge *Callyspongia* sp. collected offshore at Sanya in China [259]. Compound **952** showed both a significant inhibitory effects in the CPE inhibition assay with an IC_50_ value of 91.5 μM and a weak NF-*κ*B inhibitory effect with an IC_50_ value of 40 μM. Compounds **953**–**955** were isolated from the Caribbean marine sponge *Ectyoplasia ferox* [260]. Unexpectedly, all these saponin derivatives showed very low activity against the Jurkat and CHO cell lines in an MTT in vitro assay, as well as a hemolysis assay. Nine new triterpene galactosides and aglycones (**956**–**964**) were isolated from the tropical sponge *Lipastrotethya* sp. collected from Micronesia [261]. These compounds exhibited weak-to-no activity against the K562 human erythroleukemia cell line with IC_50_ values of 17.0, 12.5, >100, >100, 14.8, 38.9, 14.6, >100, and 27.5 μM, respectively. The chemical structures of compounds **947**–**964** are depicted in Figure 26.

Seven new triterpene glycosides (**965**–**971**) were isolated from an EtOH extract of the Caribbean sponge *Erylus goffrilleri* collected from the Caribbean Sea near the Arrecife–Seco reef [262]. Compound **967** showed moderately active hemolytic and significant cytotoxic activities, whereas **969** possessed moderate cytotoxic and low hemolytic activities. In 2020, two new triterpene galactosides, melophluosides A85 (**972**) and B86 (**973**), were successfully isolated from *Melophlus sarassinorum* collected in Siladen, North Sulawesi [263,264]. Both compounds exhibited moderate cytotoxic activity against HeLa cells with IC_50_ values of 11.6 and 9.7 mM, respectively. A simultaneous report on the same marine sponge *Clathria gombawuiensis* revealed the isolation of an additional new nortriterpene, saponin (**974**) [224]. Compound **974** exhibited moderate activity against the K562 and A549 cell lines with LC_50_ values of 2.1 and 1.8 μM, respectively, and against the bacteria *Bacillus subtilis* and *Proteus hauseri*, with MIC values of 1.6 and 3.1 μg/mL, respectively. Ten new norterpene alkaloids, coscinoderines A–J (**975**–**984**), were isolated from the marine sponge *Coscinoderma bakusi* [265]. Each coscinoderine contains a 1,2,5-trisubstituted pyridinium moiety bearing a terpene unit at the C-2 position. Mama and colleagues isolated two new sarasinosides, 5,8-epoxysarasinoside (**985**) and 8,9-epoxysarasinoside (**986**), from the marine sponge *Petrosia nigricans*, collected off the coast of Lipata, Philippines [266]. Both compounds exhibited low cytotoxicity against the HCT116 (colon) and A549 (lung) cancer cell lines. Six new triterpenoids (**987**–**992**) were isolated from the sponges *Erylus formosus* and *Penares* spp., respectively [267]. The isolated triterpenoid **992** was cytotoxic against human leukemia HL-60 cells (IC_50_ = 9.7 μM). New nor-isomalabaricanic acids, Stellettins W–X (**993**–**994**), were isolated from polar fractions of an EtOH extract of the Vietnamese sponge *Rhabdastrella globostellata*. Stellettin W was the first isomalabaricane derivative with a 13,14-epoxide group in its structure [268].

The chemical structures of compounds **965**–**994** are depicted in Figure 27, while the remaining information for compounds **931**–**994**, including names and marine sources, is presented in Appendix A.

### 3.5. Tetraterpenes

Tetraterpenoids consist of eight isoprene units and usually have a series of conjugated double bond chromophores in the molecule; therefore, these compounds usually have a color. In 2010, two new tetraterpenoids (**995**–**996**) were isolated from the Korean marine sponge *Phorbas gukulensis* [269]. They both had an unprecedented skeleton with a bis-tropolone moiety. In addition, compounds **995** and **996** exhibited significant cytotoxicity against the human pharynx, stomach, colon, and renal cancer cell lines with IC_50_ values ranging from 0.05 to 0.80 μM. Luo and coworkers isolated one polyprenylbenzaldehyde derivative (**997**) from the sponge *Luffariella variabilis* collected in the South China Sea [229]. Compound **997** demonstrated cytotoxic activity against several human cancer cell lines with IC_50_ values ranging from 3.5 to 21 μM.

The chemical structures of compounds **995**–**997** are depicted in Figure 28, while the remaining information, including names and marine sources, is presented in Appendix A.

## 4. Conclusions

Terpenes, as a broadly distributed class of MNPs within sponges, exhibit a diverse array of structures and activities, demonstrating substantial medicinal potential. This paper reports 997 sponge-derived terpenes, including sesquiterpenes, diterpenes, sesterterpenes, and triterpenes, as well as tetraterpenoids, over the past fifteen years (from January 2009 to December 2024).

Among the sesquiterpenes, the quinone/hydroquinone sesquiterpenes of the drimane (**90**–**152**, etc.), arenarol, and arenarone types (**153**–**164**, etc.) share a common rearranged drimane skeleton (**1**–**39**, etc.). The relative positions of the C-4 carbon double bond and/or the stereochemical configuration with respect to C-5 are different. The C-9 position is decorated with hydroxylated or heteroatom-substituted benzoquinone side chains. They often possess antibacterial, cytotoxic, and antioxidant activities. It is speculated that, when there are multiple hydroxyl and carboxyl groups in the structure, the active hydrogen atoms on these two types of groups can be easily released. As a result, they can react with free hydroxyl radicals (•OH), endowing them with significant antioxidant properties. Bisabolane-type sesquiterpenes (**69**–**89**, etc.) are a class of sesquiterpenes with a simple skeleton, and oxygen bridges and three-membered oxygen rings are often seen in the structure. Nitrogen-containing bisabolane-type sesquiterpenes (**69**, **78**–**79**, **81**–**83**) have only been reported from marine organisms. They often have anti-inflammatory and cytotoxic activities. For instance, the terminal ethylene group or the dienone group enhances anti-inflammatory activity. Conversely, the oxidation of the C-15 position or the methylation of the OH-7 group leads to a decline in this activity. The aromatization of the six-membered ring, as well as the oxidation of C-10/11, exerts a substantial influence on cytotoxicity.

Spongian diterpene (**321**–**367**, etc.) is a large class of diterpenes, most of which have a 6,6,6,5-tetracyclic ring system carbon skeleton and are highly oxidized, especially at the C-17 and C-19 positions and on all A–D rings. That gives them numerous biological activities, such as cytotoxicity, osteoclast inhibitory activity, and antiparasitic activity. Sesterterpenes are commonly found in the secondary metabolites of sponges. Scalarane-type sesterterpenes (**564**–**665**, etc.) often have a tetracyclic structure or a few pentacyclic structures. Compounds with hydroxyl or acetoxy groups in the C ring often have higher activity. On this basis, if the D ring also has a furan or lactone ring, it has better antitumor activity. The common triterpenes are isomalabaricane alkanes (**897**–**939**, etc.), and some triterpenes with linked carbohydrate chains (**940**–**973**, etc.). Isomalabaricane alkanes have limited structural diversity, usually with side chain variations, while the linked carbohydrate chains give the compounds better water solubility and are more amenable to bioavailability and transformation. Terpenes are widespread in nature, with an extremely diverse range of species and highly variable structures, resulting in a wide variety of biological activities. This is related not only to the structure of the compounds themselves but also to the test items selected by researchers. For example, with meroterpenes, another classification of terpenoids, this category contains a large number of compounds whose biological activities are greatly affected by structural units other than the terpene moiety, such as alkaloids and polyketides. More importantly, current research has only revealed a fraction of the biological activity of terpenes, far from all their potential activities, and a considerable portion of their biological activities remains to be discovered. Therefore, it is a huge challenge to find out the structure–activity relationship laws applicable to all terpenes.

## 5. Discussion

In contemporary research, marine natural products (MNPs) have remained a critical resource for new drug discovery. The number of studies on Porifera has been increasing worldwide, demonstrating their importance and research value [270]. Marine sponge-derived terpenes have a wide range of activities, including anti-fouling and antimalarial activities, in addition to the main activities, such as antioxidant, antibacterial, antifungal, and antitumoral activities, which are illustrated in Figure 1. Most of these compounds have only been tested for in vitro bioactivity, and the mechanism of pharmacological activity remains unclear. Consequently, further comprehensive and objective assessments must be made in conjunction with in vivo and in vitro tests. Furthermore, the pharmacokinetics, pharmacodynamics, and safety of these terpenes do not meet the requirements of drug resistance. In this context, those compounds with remarkable effects should be selected to carry out structural modification and structure–activity relationship studies, which will hopefully provide strong support for drug discovery.

In addition to terpenes, alkaloids are another major class of metabolites from marine sponges and their associated microorganisms. Alkaloids are typically a group of nitrogen-containing natural organic compounds, featuring complex structures and diverse biological activity [271]. However, in terms of quantity, they are fewer than terpenes. Only about 413 alkaloids were discovered from marine sponges during the period from 2000 to 2023 [272]. When terpenes combine with alkaloids, the demarcation is less clear. Many meroterpenes contain both the structure of terpenoids and the parts of alkaloids [49,62,162,168,235,241,265]. Among the various compounds, meroterpenes are large in number and diverse in activity. The classification and identification of meroterpenes are not easy [273,274,275]. Therefore, the classification in this paper is broadly based on the number of isoprenoids, the structural unit of the terpenes, and the number of carbons. In conclusion, these marine natural products all exhibit remarkable activities and can be used, either directly or indirectly, as the lead compounds for derivatization synthesis [276].

On the other hand, traditional separation methods, such as the OSMAC (one strain many compounds) strategy, are no longer sufficient for today’s separation needs, because they are time-consuming and mostly purposeless, making them inefficient when used alone. As a matter of fact, the boundaries between disciplines have been gradually blurred in the continuous development of scientific research. More often than not, we need interdisciplinary co-operation in order to achieve our research goals with less effort. For example, the creation of databases by collecting the key annotation information and contributions of known MNPs can provide an important database for drug mining and screening [277]. The use of MS/MS-based molecular networking (GNPS) to guide extract selection is growing rapidly, and many papers have specifically used the technique to select extracts for investigation [53,278]. Traditional biology mainly takes living organisms in the natural world as research objects; applies scientific methods such as observation, experiment, analysis, and reasoning; and adopts a variety of technological means to obtain and process a large amount of biological data, establishing and verifying biological models and theories with the main goal of exploring the nature and laws of life. The development of technology makes it possible to carry out systematic engineering modifications of living organisms [279]. Synthetic biology is an emerging intersection of the biological and chemical disciplines, with the aim of creating, controlling, and reprogramming the artificial biosystems that operate like electrical circuits [280,281]. It is possible to obtain a certain component in a targeted manner or to increase its yield by this means, solving the disadvantage of the low yield of MNPs.

It is encouraging to see that there is sustained interest in innovative drug discovery and development based on MNPs. In future research, interdisciplinary collaboration between chemical synthesis, biosynthesis, bioactivities, ecological studies, genomic studies, and other related fields should be emphasized to enhance the success rate of new drug discovery. Social and scientific progress follows the Cannikin law that no discipline can exist separately from all others. The development of human civilization requires the combined efforts of all our disciplines and all their researchers.

## Figures and Tables

**Figure 1 marinedrugs-23-00096-f001:**
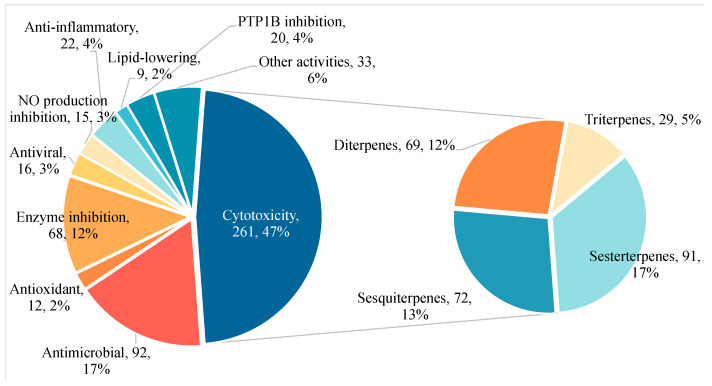
Percentage distribution of new compounds with different bioactivities for 2009–2024.

**Figure 2 marinedrugs-23-00096-f002:**
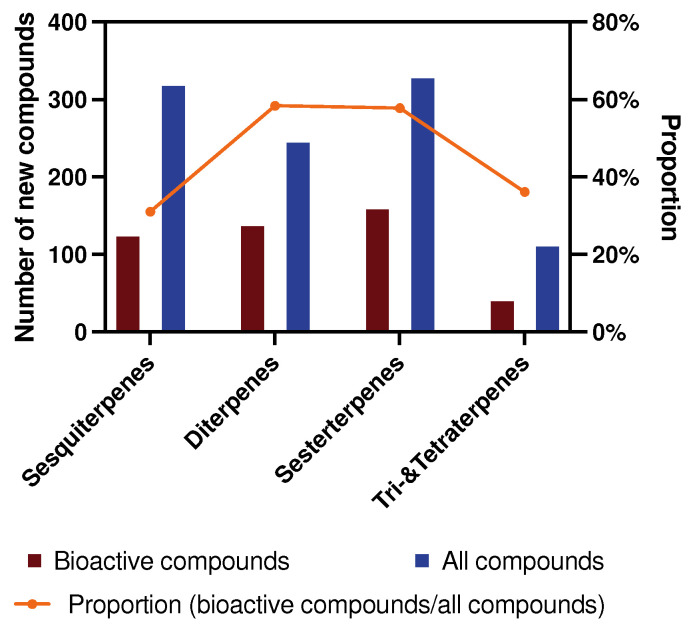
The number and proportion of new bioactive compounds in different terpenes for 2009–2024.

**Figure 3 marinedrugs-23-00096-f003:**
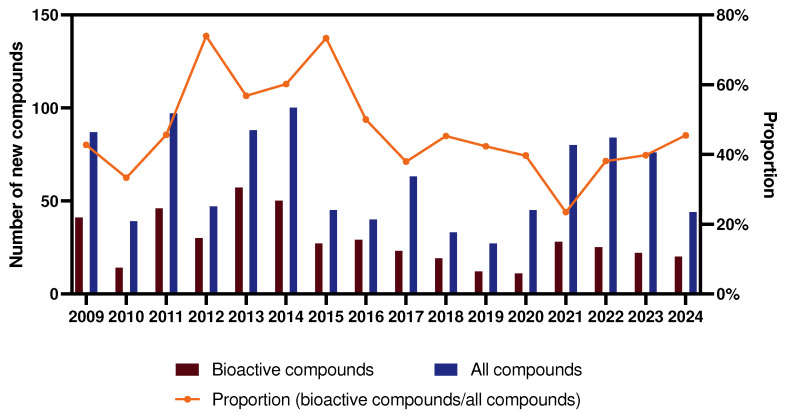
Trends in the number and proportion of new bioactive compounds for 2009–2024.

**Figure 4 marinedrugs-23-00096-f004:**
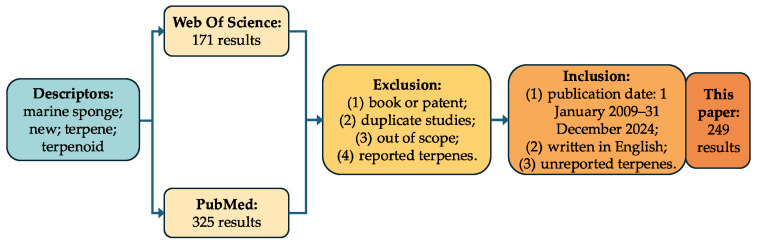
The data collection flowchart of this research.

**Figure 5 marinedrugs-23-00096-f005:**
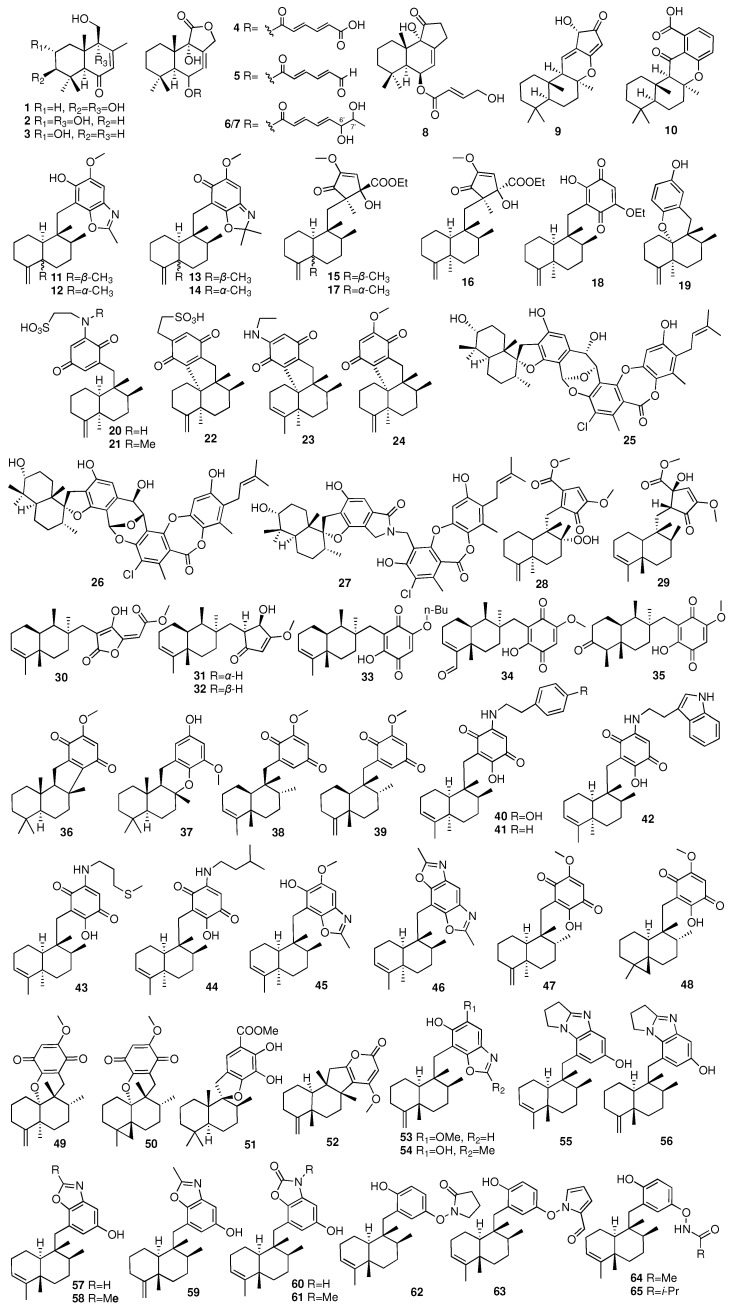
Chemical structures of compounds **1**–**65**.

**Figure 6 marinedrugs-23-00096-f006:**
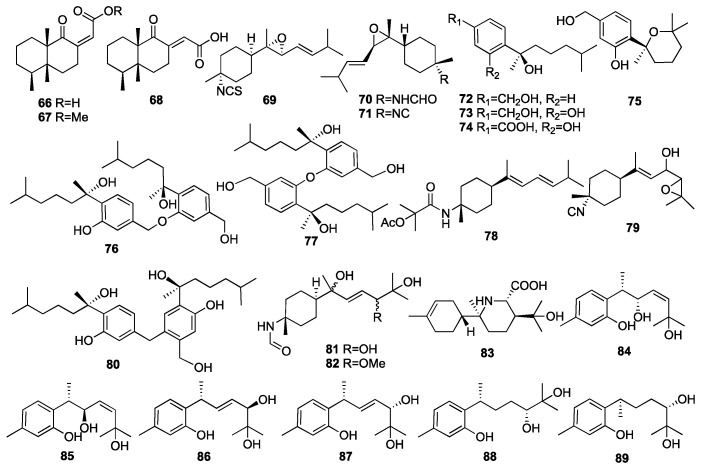
Chemical structures of compounds **66**–**89**.

**Figure 7 marinedrugs-23-00096-f007:**
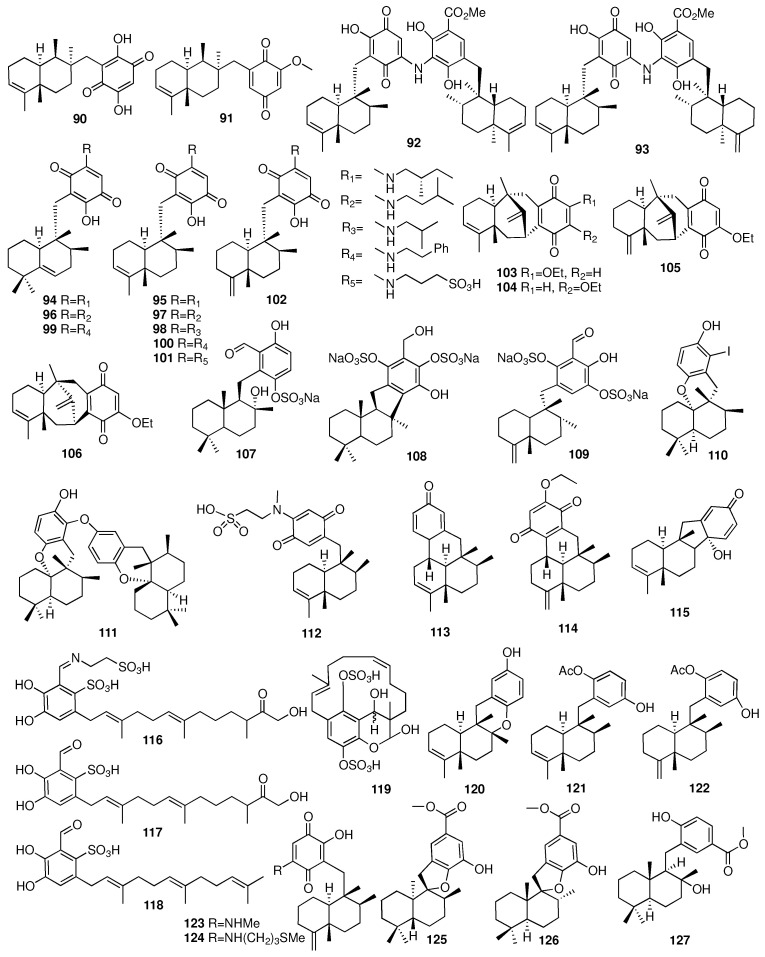
Chemical structures of compounds **90**–**127**.

**Figure 8 marinedrugs-23-00096-f008:**
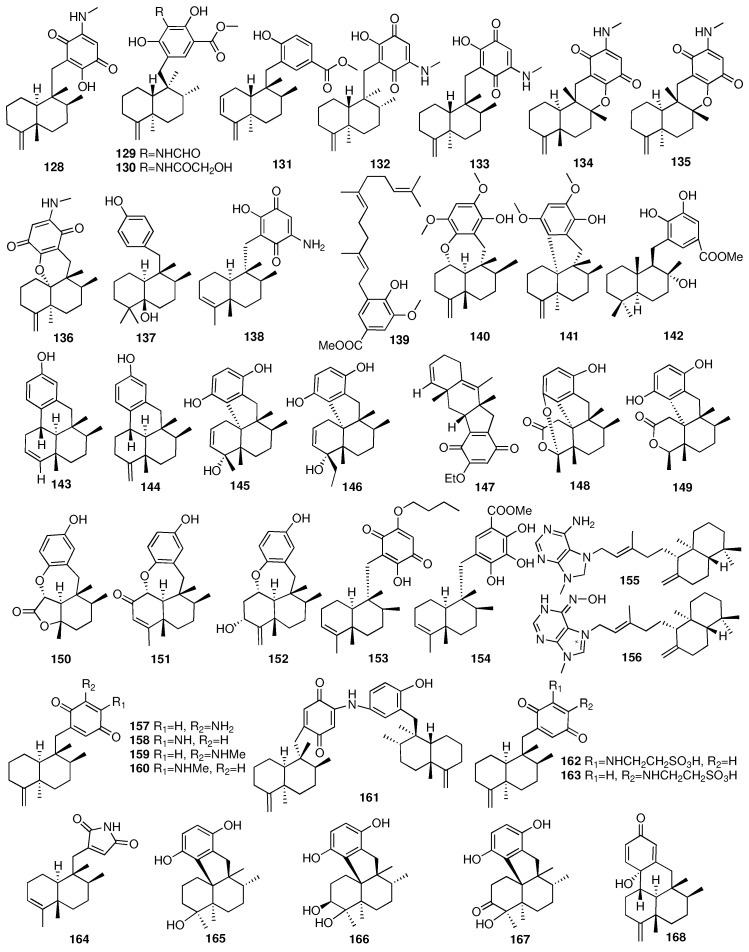
Chemical structures of compounds **128**–**168**.

**Figure 9 marinedrugs-23-00096-f009:**
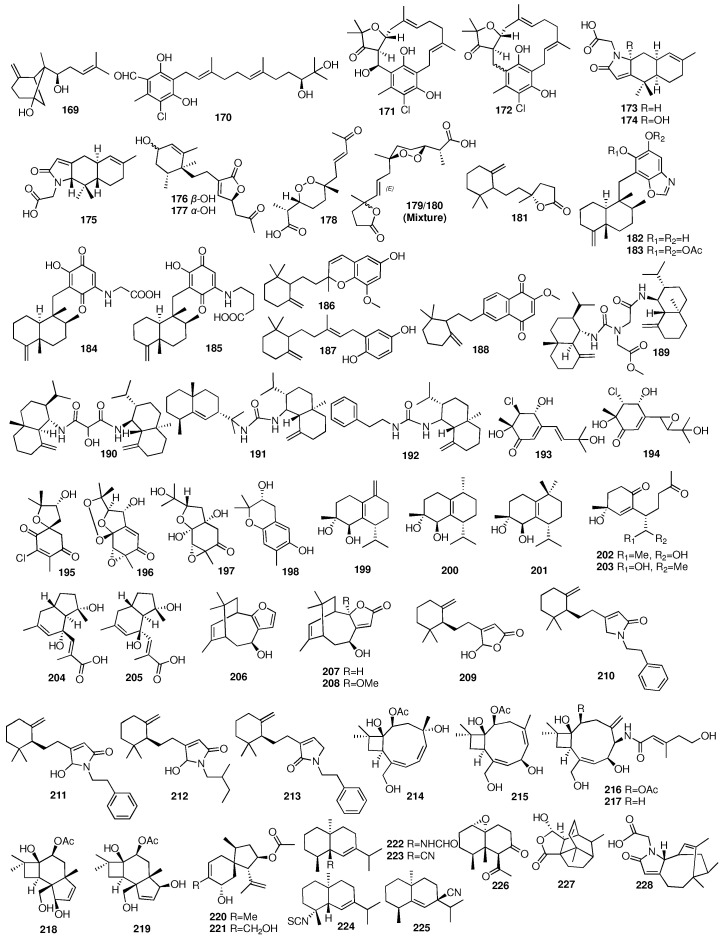
Chemical structures of compounds **169**–**228**.

**Figure 10 marinedrugs-23-00096-f010:**
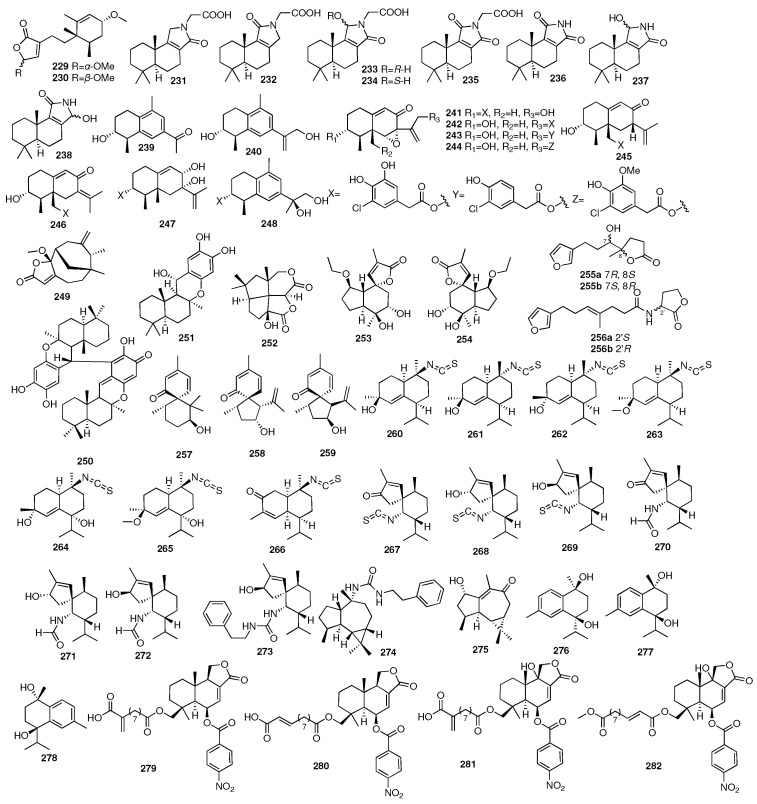
Chemical structures of compounds **229**–**282**.

**Figure 11 marinedrugs-23-00096-f011:**
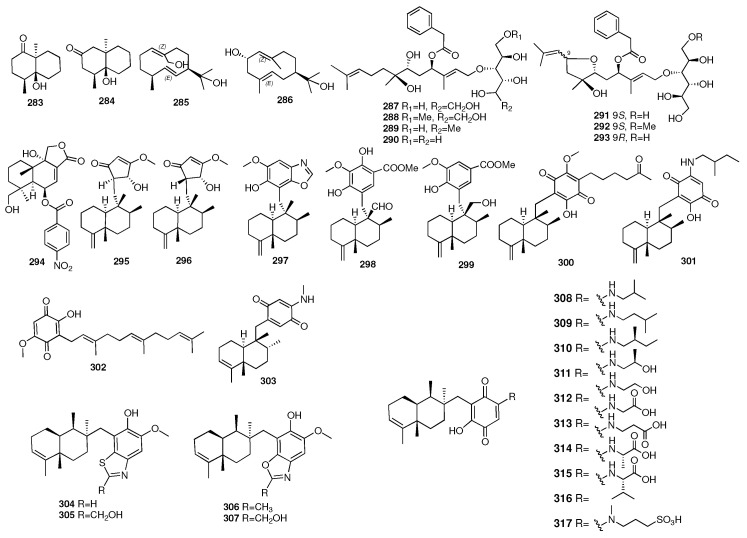
Chemical structures of compounds **283**–**317**.

**Figure 12 marinedrugs-23-00096-f012:**
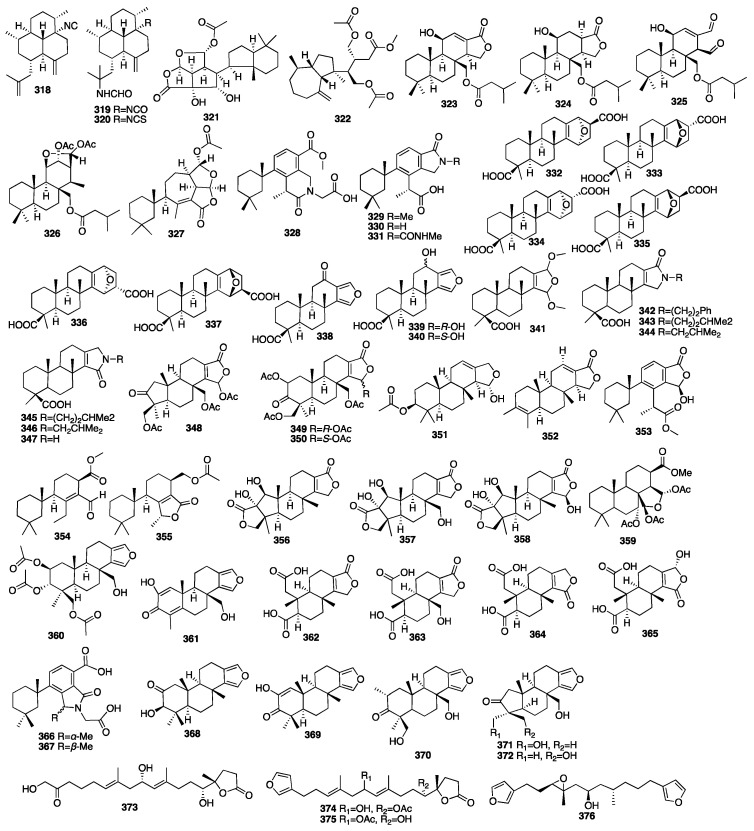
Chemical structures of compounds **318**–**376**.

**Figure 13 marinedrugs-23-00096-f013:**
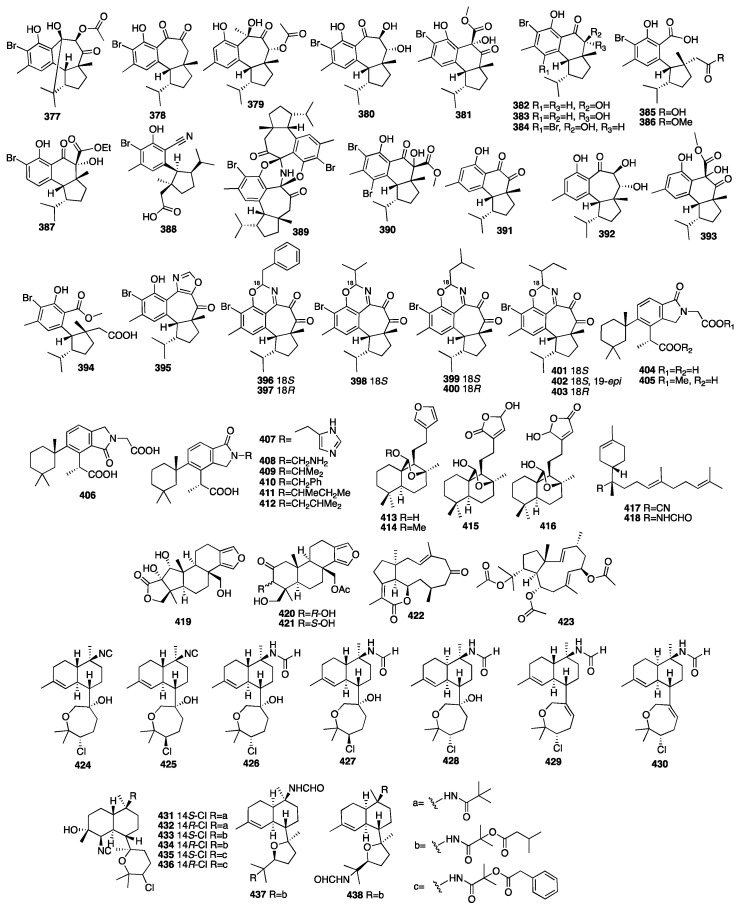
Chemical structures of compounds **377**–**438**.

**Figure 14 marinedrugs-23-00096-f014:**
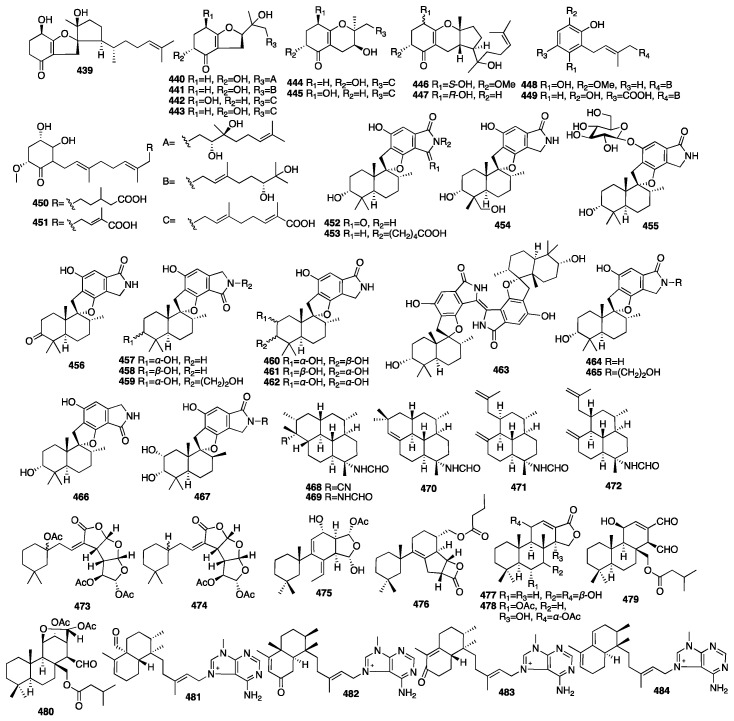
Chemical structures of compounds **439**–**484**.

**Figure 15 marinedrugs-23-00096-f015:**
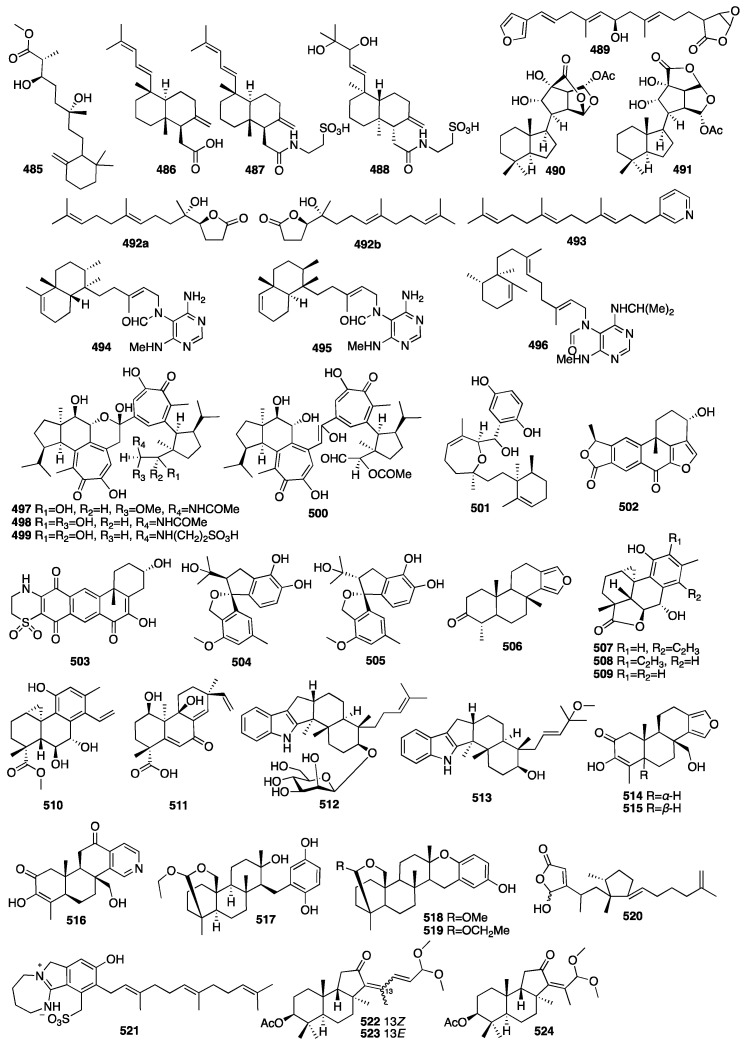
Chemical structures of compounds **485**–**524**.

**Figure 16 marinedrugs-23-00096-f016:**
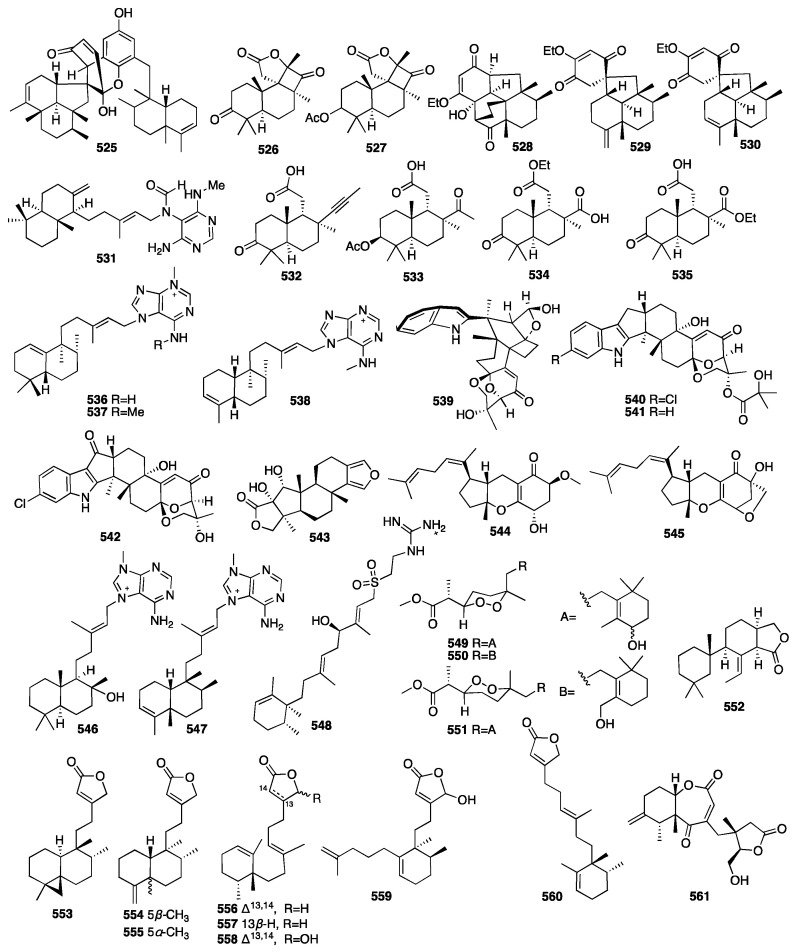
Chemical structures of compounds **525**–**561**.

**Figure 17 marinedrugs-23-00096-f017:**
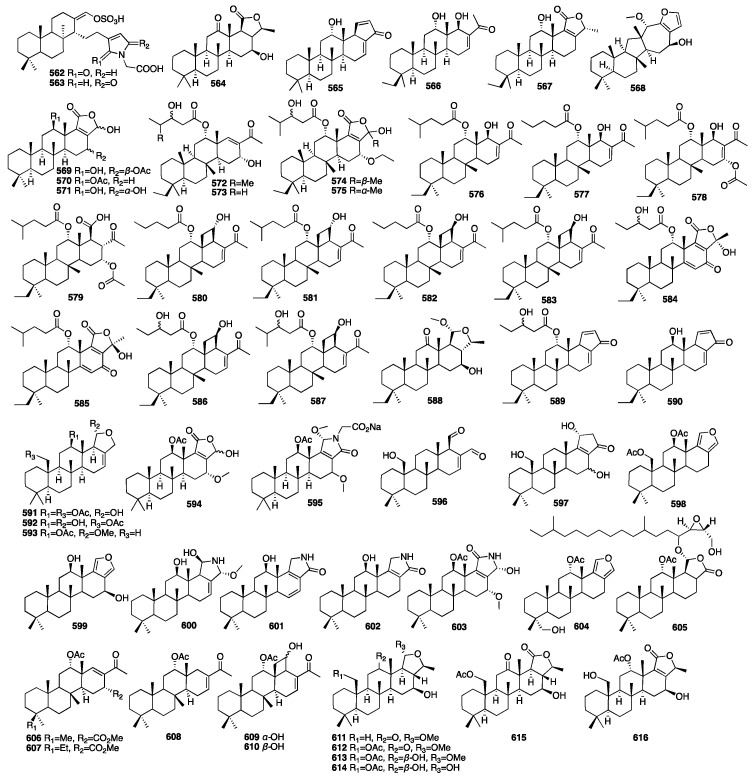
Chemical structures of compounds **562**–**616**.

**Figure 18 marinedrugs-23-00096-f018:**
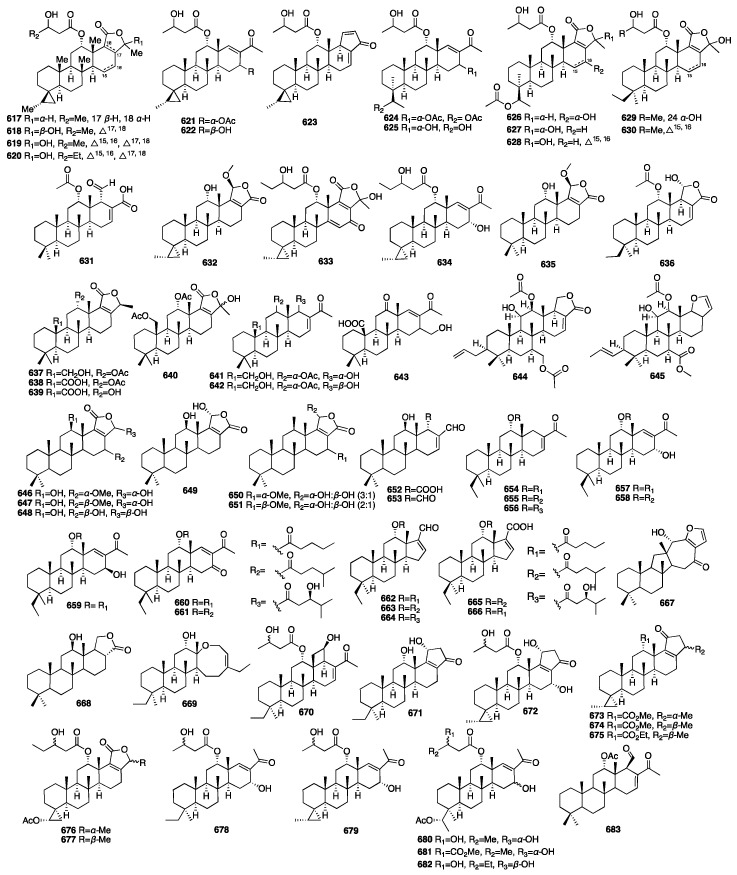
Chemical structures of compounds **617**–**683**.

**Figure 19 marinedrugs-23-00096-f019:**
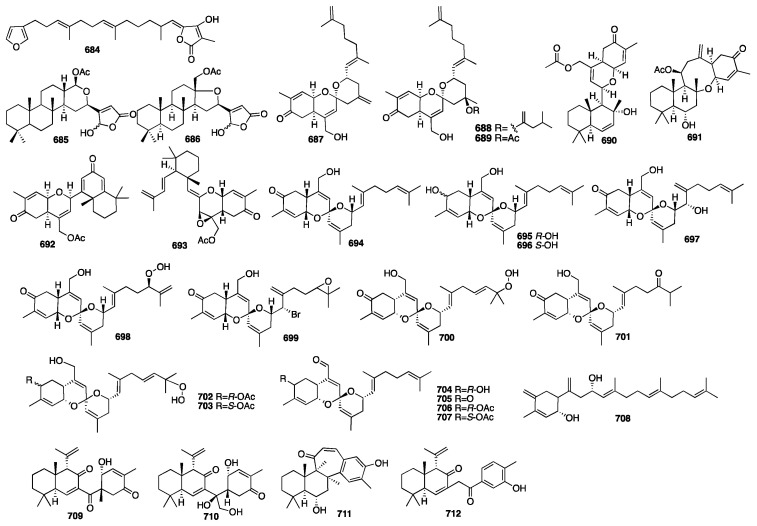
Chemical structures of compounds **684**–**712**.

**Figure 20 marinedrugs-23-00096-f020:**
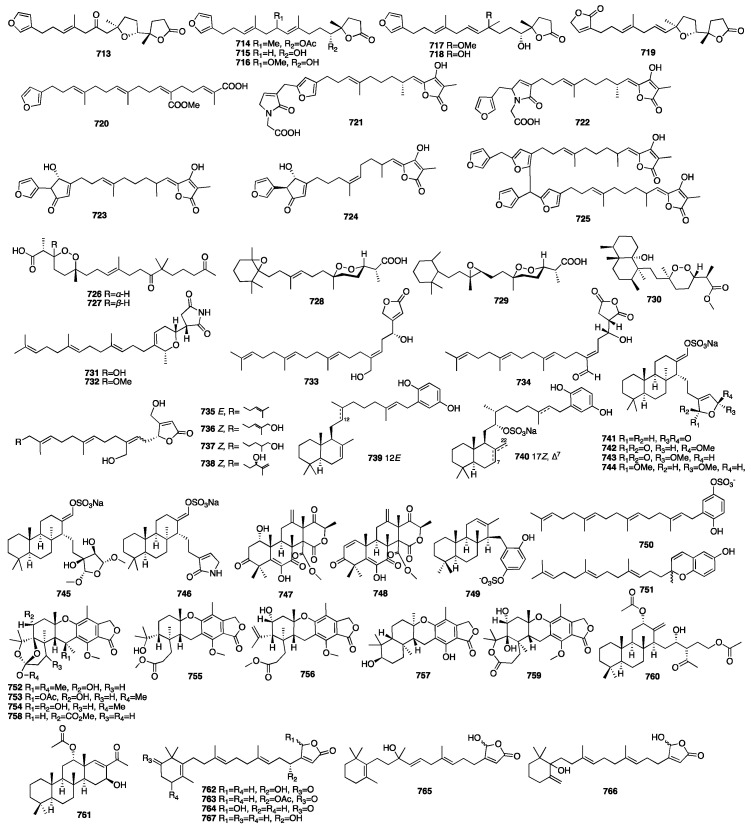
Chemical structures of compounds **713**–**767**.

**Figure 21 marinedrugs-23-00096-f021:**
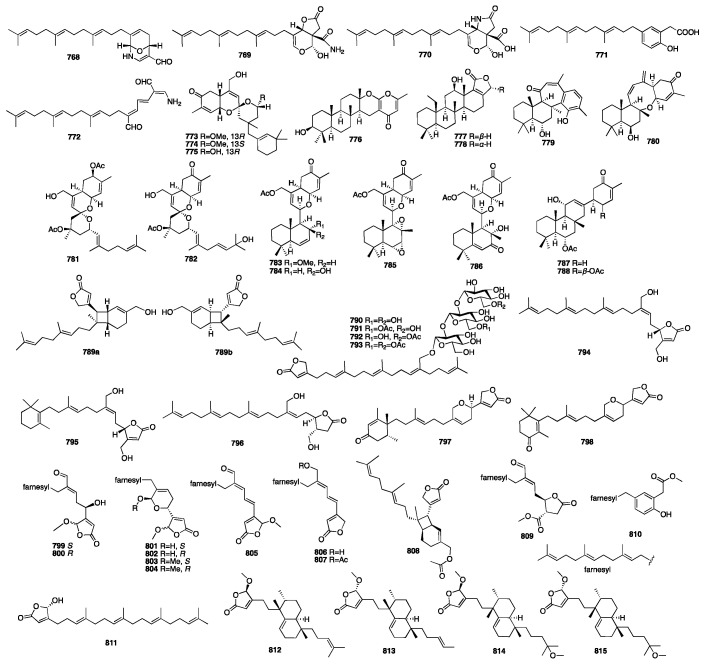
Chemical structures of compounds **768**–**815**.

**Figure 22 marinedrugs-23-00096-f022:**
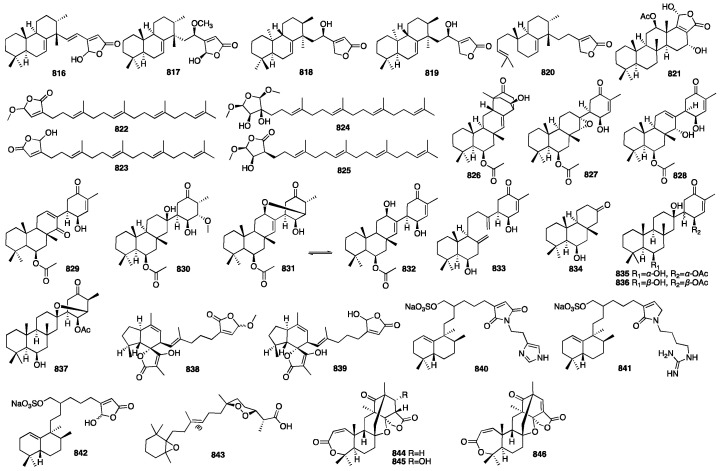
Chemical structures of compounds **816**–**846**.

**Figure 23 marinedrugs-23-00096-f023:**
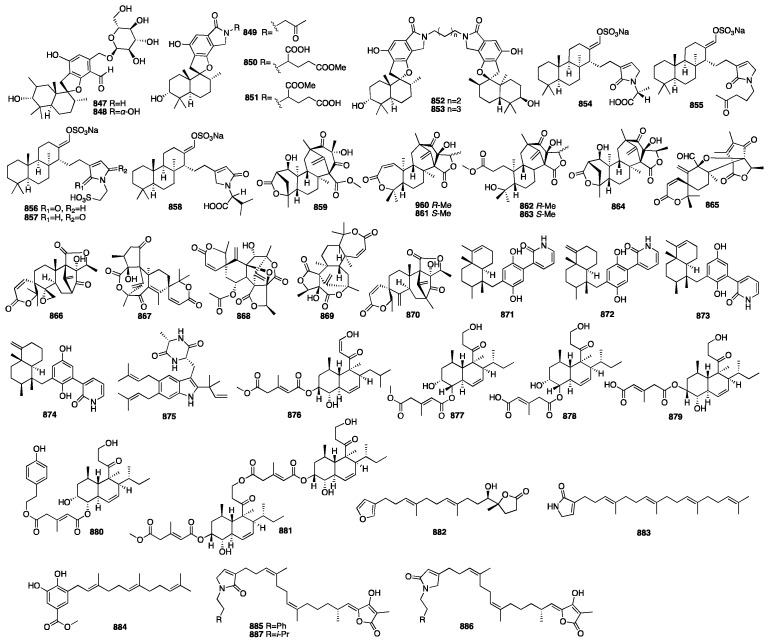
Chemical structures of compounds **847**–**887**.

**Figure 24 marinedrugs-23-00096-f024:**
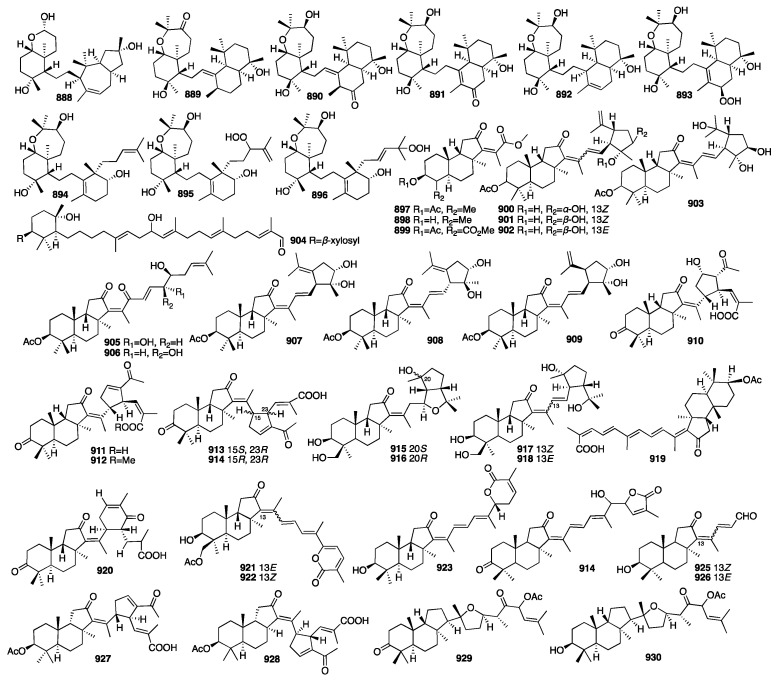
Chemical structures of compounds **888**–**930**.

**Figure 25 marinedrugs-23-00096-f025:**
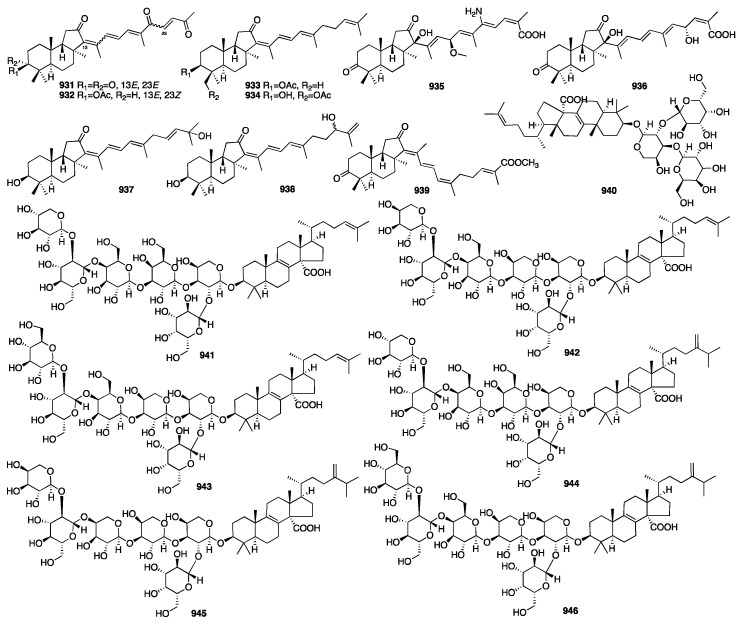
Chemical structures of compounds **931**–**946**.

**Figure 26 marinedrugs-23-00096-f026:**
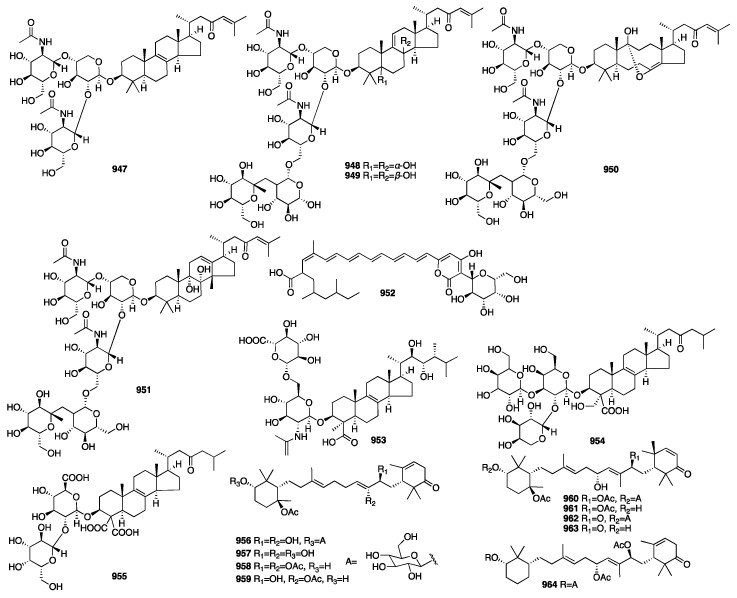
Chemical structures of compounds **947**–**964**.

**Figure 27 marinedrugs-23-00096-f027:**
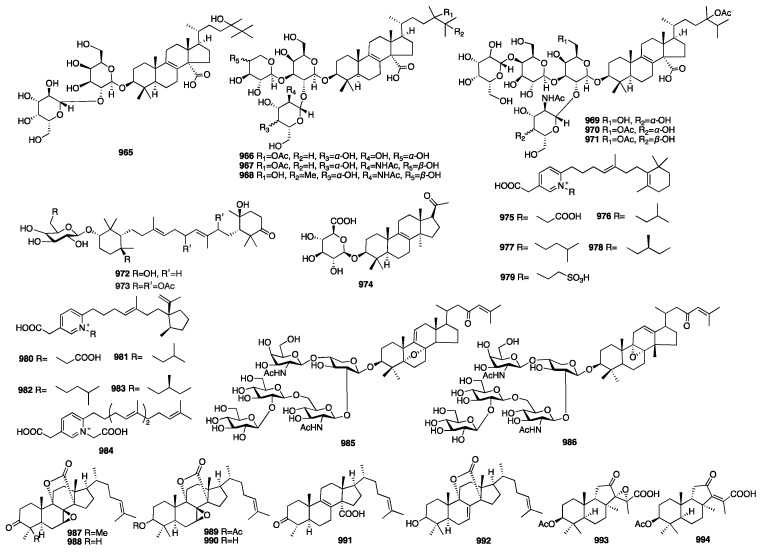
Chemical structures of compounds **965**–**994**.

**Figure 28 marinedrugs-23-00096-f028:**
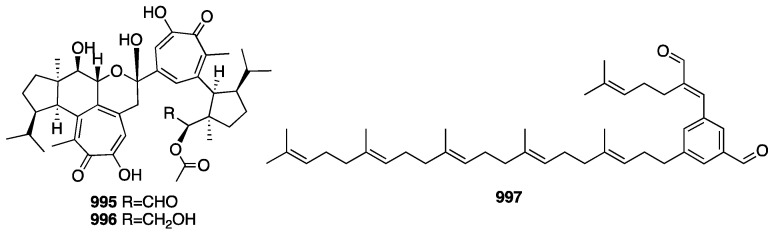
Chemical structures of compounds **995**–**997**.

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
