# Peer review of "Bioactive Terpenes from Marine Sponges and Their Associated Organisms"

_marinedrugs, 2025, doi:10.3390/md23030096_

Round 1
Reviewer 1 Report (New Reviewer)
Comments and Suggestions for Authors
This review article presents a comprehensive analysis of sponge-derived terpenes, summarizing their chemical diversity, biological activities, and potential pharmaceutical applications. Covering studies from January 2009 to December 2024, the manuscript compiles 997 novel terpene metabolites identified from 249 publications. The review highlights key biological activities, including anticancer, antimicrobial, anti-inflammatory, antioxidant, and enzyme inhibition effects.
- I suggest to add a brief paragraph about the other class of marine-derived compounds, as alkaloids, citing some references: a) Mar. Drugs 2023, 21, 412. https://doi.org/10.3390/md21070412; b) European Journal of Medicinal Chemistry Reports, 2024,100158, https://doi.org/10.1016/j.ejmcr.2024.100158.
- The manuscript is largely descriptive, providing a catalog of compounds without deeper analysis. I suggest include a section on structure-activity relationships (SAR) to explain why certain terpenes exhibit specific bioactivities.
- I suggest adding other keywords to facilitate online searches.
- In the discussion, the authors could add a paragraph about the microbial contribution to sponge-derived terpene production.
- Include recent clinical trials or preclinical studies on marine-derived terpenes.
- Lines 405: value should be values
- In vivo and in vitro should be in italics
- The large number of tables in the main manuscript disrupts the flow of the text, making it less reader-friendly. To improve readability and maintain focus on the key findings, I recommend moving the detailed tables to the Supporting Information (SI).
- Please check all references. The doi is fully recommended.
Comments on the Quality of English Language
A language revision is recommended to improve readability and scientific clarity.
Author Response
Please see the attachment.

Reviewer 2 Report (New Reviewer)
Comments and Suggestions for Authors
The manuscript is a review on biologically active terpenes isolated from marine sponge and associated organisms and published from 2009 to 2024. The review covers very impressive array of structural data about terpenes and their derivatives possessing anti-inflammatory, antibacterial, cytotoxic, anticancer, and antioxidant properties and included 997 novel terpene metabolites. The article seems to be normal comprehensive review with identification reference numbers for each compound noted in the text, all the structures are presented at the figures, the tables present the substances names, numbers, source, biological activities, and references on the literature. The article seems to be well written however minor improvements are necessary.
1) Line 176. Replace “wae isolated” with “was isolated”.
2) Line 334. Replace “Xesto-spongia” with “Xestospongia”.
3) Use, please, the ending “en” only for single double bond and “ene” for plural numbers of double bonds, For example: buten, but butadiene.
4) Lines 504–506. Start, please, the word “compounds” from small letters but not capital.
5) Line 540. Replace “Hentschel” with “Hentschel”, i.e. no use italic for an author name.
6) Line 586. Replace, please, “Dysidea Arenaria” “Dysidea arenaria”
7) The tables 1, 2 and 3 seems to be not accurate and not full clear. What is “Sponge Strains”. Replace with “Sponge Species” or “Sponge”. The lines are not concurred in the different columns. I recommend never use empty places in the tables. Use, please, a dash in the tables as “the same” only but in alternative cases use No data, No applicable or something similar. Think, please, again about improving the tables.
8) Line 657. Replace “Compound 579” with “compound 579”.
9) Line 658. Replace “Balanus Amphitrite” with “Balanus amphitrite”.
10) Line 732. Replace “Ansellone A (689)” with “ansellone A (689)”.
11) Line 739. Replace “sp” with “sp.”
12) Page 36. Table 3. Replace “Dysidea Arenaria” with “Dysidea arenaria”.
13) Line 855. Replace “Cmccftla” with “Cmccftla”, i.e. no use italic for an author name.
14) Line 877. Replace “Lendenfeld” with “Lendenfeld”, i.e. no use italic for an author name.
15) Line 963. The phrase “combined with sugars carbohydrate” is senseless. I do not understand what is a “sugars carbohydrate”. Replace, please, with “combined with one or several carbohydrate chains”.
17) Equalize, please, the different styles in the references. Use, please, only small letters for beginning of the words (except of special cases) within the titles, or use, please only capital letters for such purposes except of prepositions, of course.
18) Use, please, correctly the long dashes. Use them between the numbers in numberic intervals, such as 100–101 without spaces or between the letters in numberic-like intervals such as A–D. Never use, please, bold style for the long dashes.
The article is interesting for the readers and may be published after minor revisions.
Author Response
Please see the attachment.

Reviewer 3 Report (New Reviewer)
Comments and Suggestions for Authors
This review addresses the implications of bioactive terpenes from marine sponges and their associated organisms. While the topic is relevant, the text (62 pages of main content) is overly lengthy, which dilutes the essence of the review. If it remains in its current form, it will be very difficult for it to pass as a narrative review due to its excessively long length. It would be advisable to target the focus more narrowly and present only the most important and novel findings. Several deficiencies in content and form were identified and should be addressed based on the following recommendations:
- The conclusion section of the abstract should be improved to better highlight the study's outcomes and future research directions.
- Multiple bibliographic citations (e.g., [5-9]) create redundancy and diminish the clarity of information. Consider specifying the references or reducing the number to make the citations more focused.
- It would be more effective to introduce the first three figures right before the respective figure in the text, rather than all at once. This will ensure a smoother flow and better clarity.
- The methodology section is poorly presented. If no further conclusive information can be added, it may be more effective to include a diagram or flowchart of the process.
- Sections L83-131; L136-168; L171-215; L218-L262; L265-314; L317-L356 consist of long paragraphs, reducing readability and comprehension. These should be reorganized into shorter, more logical paragraphs for better understanding.
- Tables 1, 2, etc. – there is no need to include the first author in the last column; just the bibliographic reference [x] should suffice. Tables should also be alphabetized by compound or species.
- The conclusions section should strictly summarize the important findings from this study and avoid referencing other works, focusing on the outcomes obtained in the current review.
Round 2
Reviewer 1 Report (New Reviewer)
Comments and Suggestions for Authors
I am satisfied with the revised version of the manuscript and recommend proceeding with publication of the manuscript
Author Response
Thank you for your encouragement and recognition. We will continue to work hard.
Reviewer 3 Report (New Reviewer)
Comments and Suggestions for Authors
The authors have significantly improved the manuscript based on the suggestions received.
One minor comment:
It is advisable to make the conclusion section separately, without references, targeting only the most important outcomes.
Author Response
Thank you for your encouragement and recognition, and we have made corresponding revisions to the manuscript.
This manuscript is a resubmission of an earlier submission. The following is a list of the peer review reports and author responses from that submission.
Round 1
Reviewer 1 Report
Comments and Suggestions for Authors
A review of the terpenes described in sponges or sponge-associated organisms is presented in their paper.
These compounds are classified into 6 groups, namely sesquiterpenes, diterpenes, sesterterpenes, triterpenes, tetraterpenes and meroterpenes, with the latter accounting for only 3.42% of the total number of compounds reported.
However, to the surprise of this reviewer, the first 7 structures (compounds 1-7, Figure 4) included in the monoterpenes section are actually meroterpenes. Similarly, in the next figure (Figure 5) which is supposed to describe sesquiterpenes, again a considerable number of meroterpenes appear, and even a sterol (compound 25).
This situation is repeated again in Figure 6, Figure 7, Figure 8 and so on.
Although I recognize that sometimes the correct identification of meroterpenes can be problematic (Nat. Prod. Rep., 2023, 40, 1071-1077), the incorrect classification that is spread throughout the article becomes even more difficult for this reviewer to understand if one takes into account that in several of the original papers these compounds are indeed described as meroterpenes.
This leads me to conclude that the article cannot be accepted for publication.
If the authors intend to resubmit their work to this or another journal, the following considerations could, in the opinion of this reviewer, contribute to the quality of the review:
Once the compounds described have been isolated from sponges or sponge-associated organisms, this data should be included in the title of the article.
A description of the literature search strategy would be helpful. How, where and when did you search for the information: databases, keywords, etc.
-The authors should check the structures of the compounds included in this review. For example, the structure of compound 32 is clearly incorrect.
Once the compounds are correctly classified, the compounds should be grouped within each section according to their skeleton.
Since the new classification will mean that the number of compounds in the meroterpenes section will increase significantly, it may be appropriate to include some revision of meroterpenes (Mar. Drugs 2013, 11, 1602; Biomolecules 2021, 11, 957....). In this regard, depicting the terpene scaffold and the elements of different biosynthetic origin in the corresponding meroterpene with different colors would be highly appreciated.
Reviewer 2 Report
Comments and Suggestions for Authors
This paper provides a well-structured summary of 1,044 novel terpene compounds extracted from marine sponges over the past 15 years. The detailed classification by type of biological activity, as well as the breakdown by species and source of sponges, is particularly useful. However, the review could be further enhanced by addressing several points outlined below.
1. Overall, there are numerous grammatical errors and typos throughout the paper. A few examples found on the first page are as follows:
Line 10: "remain" → "have remained"
Line 12: "producer" → "producers"
Line 17: "1044" → "1,044", "terpenes metabolites" → "terpene metabolites"
Line 21: "systematical" → "systematic"
Line 30: "high-pressured" → "high-pressure"
Line 30: "anoxgenous" → "anoxic"
Line 34: "Crytothya Cryta" → "Cryptotheca crypta"
Line 37: "Large number of natural compounds" → "A large number of natural compounds"
It would be advisable to review and correct the English grammar throughout the paper.
2. Lines 28–30: The statement, “After more than 500 million years of evolution, marine sponges have developed a unique chemical and physical defense mechanism to adapt to the high-pressured, high-salted, anoxgenous, and lucifugal environment,” requires a reference.
3. References 4–13 appear to be annual marine natural product reviews. It would be better to either cite only the most recent review or arrange the references chronologically if including them all.
4. Figure 1a seems unnecessary for this review article. The number of papers published per country is less relevant here. Instead, a graph illustrating the number of studies by sponge genus or order, or by activity type, would be more valuable.
5. Line 74: "Negombata corticate" should be corrected to "Negombata corticata."
6. Tables 1–5 should be referenced in the text, but they are currently not mentioned. Additionally, some sponge names are incorrectly labeled in Table 1. Compound names are also inconsistently abbreviated as "compound + number"; they should all be replaced with actual names.
7. "μg·mL-1" should be changed to "μg/mL."
8. In Figure 5, the five-membered rings in the moieties of compounds 17–19 are distorted and should be redrawn.
9. For compounds 44 and 45 in Figure 5, the differences between the compounds should be clearly indicated.
10. Hyphens ("-") between page numbers in references should be replaced with en-dashes ("–").
Addressing these points could greatly improve the clarity and quality of the review.
Comments on the Quality of English Language
I pointed out some grammatical errors in the comments above, so please reflect them.
Round 2
Reviewer 1 Report
Comments and Suggestions for Authors
In my first review, my main objection to this work was the incorrect classification of structures, especially the classification of compounds with meroterpene structure.
In this sense, and although I must recognize that the authors have made an effort to reclassify compounds (the substances classified as meroterpenes have increased from about 25 to more than 300), the inaccuracies and errors that still persist in the revised version of the article go beyond what is acceptable and reasonable.
Also really striking are the errors in the description of the references.
Notice the following examples:
-Compound 656, the second “meroterpene” reported is actually a steroid.
-Compounds 687-692, also clasiffied as meroterpnes, were described in the original article as “Norterpenoids and Related Peroxides …” Furthermore, the reference given for these compounds in Table 5 is wrong.
-Compounds 727-729 are bisabolane derivatives as clearly established by the original authors in the title of their publication. “New bisabolane sesquiterpenoids from…”. Again, the reference given for these compounds in Table 5 is wrong.
-Compound 736 is also wrongly classified as meroterpene. This misclassification is especially striking when the original authors of the corresponding research titled their work as: “Bicyclic C21 terpenoids from the marine...”. Regarding the corresponding literature reference, this appears in the main text as 193, in Table 5 as 198, whereas the actual reference is 196.
-Something very similar was found with compounds 749-752. Although classified as meroterpenes, the original paper was titled: “Cytotoxic Diterpenoid Pseudodimers from…”. Regardind the reference, this appears in the main text as 197, in Table 5 as 202, whereas the actual reference is 200.
- The very same errors in the classification and the lack of accuracy in the numbering of the references can be found for, among others, compounds 753-754, 830-831, 896-899, 927-928, 948-950, 982-987.
- In addition to the numerous errors found in the meroterpenes part, other compounds have been erroneously classified, such as the allegedly "triterpene" 596.
-Apart from the classification mistakes, authors did not follow this reviewer's prior recommendation; “Once the compounds are correctly classified, the compounds should be grouped within each section according to their skeleton”. In my opinion, the fact that all the structures are mixed (especially in the case of the meroterpenes) makes it significantly more difficult to read and understand the content of the review.
All this makes it impossible for me to recommend this article for publication.
Finally, I would suggest the authors to consider the assistance of a specialist in the classification and biosynthesis of natural products.
Reviewer 2 Report
Comments and Suggestions for Authors
The authors have faithfully incorporated the reviewers' comments in the major revision. Therefore, this paper can be accepted for publication.